# Metropolis-Hastings Data Augmentation for Graph Neural Networks

**Hyeonjin Park**[1][*], **Seunghun Lee**[1][*], **Sihyeon Kim**[1], **Jinyoung Park**[1]
**Jisu Jeong**[2,3], **Kyung-Min Kim**[2,3], **Jung-Woo Ha**[2,3], **Hyunwoo J. Kim**[1][†]
Korea University[1], NAVER CLOVA[2], NAVER AI LAB[3]
{hyeonjin961030, llsshh319, sh_bs15, lpmn678, hyunwoojkim}@korea.ac.kr
{jisu.jeong, kyungmin.kim.ml, jungwoo.ha}@navercorp.com

## Abstract

Graph Neural Networks (GNNs) often suffer from weak-generalization due to sparsely labeled data despite their promising results on various graph-based tasks. Data augmentation is a prevalent remedy to improve the generalization ability of models in many domains. However, due to the non-Euclidean nature of data space and the dependencies between samples, designing effective augmentation on graphs is challenging. In this paper, we propose a novel framework Metropolis-Hastings Data Augmentation (MH-Aug) that draws augmented graphs from an explicit target distribution for semi-supervised learning. MH-Aug produces a sequence of augmented graphs from the target distribution enables flexible control of the strength and diversity of augmentation. Since the direct sampling from the complex target distribution is challenging, we adopt the Metropolis-Hastings algorithm to obtain the augmented samples. We also propose a simple and effective semi-supervised learning strategy with generated samples from MH-Aug. Our extensive experiments demonstrate that MH-Aug can generate a sequence of samples according to the target distribution to significantly improve the performance of GNNs.

## 1 Introduction

Graph Neural Networks (GNNs) [1] have been widely used for representation learning on graph-structured data due to their superior performance in various applications such as node classification [2–4], link prediction [5–7] and graph classification [8, 9]. They have been proven effective by achieving impressive performance for diverse datasets such as social networks [10], citation networks [4], physics [11], molecules [12], and knowledge graphs [10]. However, GNNs often suffer from weak-generalization due to their small and sparsely labeled graph datasets. One prevalent remedy to address the problem is data augmentation. Data augmentation increases the diversity of data and improves the generalization power of machine learning models trained on randomly augmented samples. It is widely used to enhance the generalization ability of models in many domains. For instance, in image recognition, advanced methods like [13–15] as well as simple transformations such as random cropping, cutout, Gaussian noise, or blurring have been used to achieve competitive performance.

However, unlike image recognition, designing effective and label-preserving data augmentation for individual samples on graphs is challenging due to their non-Euclidean nature and the dependencies between data samples. In image recognition, it is straightforward to identify operations that preserve labels. For instance, human can verify that rotation, translation, and small color jittering do not change the labels in image classification. In contrast, graphs are less interpretable and it is non-trivial

---

[*]First two authors have equal contribution

[†]is the corresponding author

for even human to check whether the augmented samples belong to the original class or not. In addition, due to the dependencies between nodes and edges in a graph, it is hard to control the degree of augmentation for individual samples. For instance, a simple operation on a graph, *e.g.*, dropping a node, may result in a completely different degree of augmentation depending on the graph structure. If a hub node is removed, the single perturbation affects a substantial amount of other nodes, which are data samples in node classification. To address these challenges, learning-based data augmentation methods for graphs have been proposed. AdaEdge [16] optimizes the graph topology based on the model prediction. [17] proposes GAug-M and GAug-O that generate augmented graphs via a differentiable edge predictor. GraphMix [18] presents interpolation-based regularization by jointly train a fully connected network and graph neural networks. However, they require additional models for augmentation and more importantly do not *explicitly* guarantee that augmentation has a proper strength and diversity.

In this paper, we proposed a novel framework called Metropolis-Hastings Data Augmentation (MH-Aug) that draws augmented graphs from an '*explicit*' target distribution with the desired strength and diversity for semi-supervised learning. Since the direct sampling from the complex distribution is challenging, we adopt the Metropolis-Hastings algorithm to obtain the augmented samples. Recently, the importance of leveraging unlabeled data as well as adopting advanced augmentation has emerged [19, 20]. Inspired by that, we also adopt the consistency training by utilizing the regularizers for unlabeled data. Our extensive experiments demonstrate that MH-Aug can generate a sequence of samples according to the desired distribution and be combined with the consistency training and it significantly improves the performance of graph neural networks.

Our **contributions** are summarized as follows:

- We proposed a novel framework Metropolis-Hastings Data Augmentation that draws augmented samples from an 'explicit' target distribution. To the best of our knowledge, this is the first work that studies data augmentation for graph-structured data from a perspective of a Markov chain Monte Carlo sampling.
- We *theoretically* and *experimentally* prove that our MH-Aug generates the augmented samples according to the desired distribution with respect to the strength and diversity.
- We propose a target distribution that flexibly controls the strength and diversity of augmentation. This includes an efficient way to measure the strength of augmentation reflecting the structural changes of ego-graphs (or samples in node classification).
- Lastly, we propose a simple and effective semi-supervised learning strategy leveraging sequentially generated samples from our method.

## 2  Related Works

**Semi-Supervised Learning on Graphs.** GNNs have been widely adopted in representation learning on graphs [2–4]. However, existing works only utilize a small subset of nodes. To fully utilize a large amount of unlabeled data, recent studies for semi-supervised learning have emerged inspired by semi-supervised frameworks in other domains [19, 20]. GraphMix [18] is a regularization method based on semi-supervised learning by linear interpolation between two data on graphs, and SSL [21] proposes self-supervised learning strategies to exploit available information from graph structure. BVAT [22] promotes the smoothness of GNNs by generating virtual adversarial perturbations. Likewise, we follow semi-supervised strategy to leverage unlabeled data while considering sequentially generated samples from our augmentation.

**Data Augmentation on Graphs.** Data augmentation is an effective technique to improve generalization by increasing the diversity of data. It is becoming the *de facto* necessity for model training to employ simple data augmentation (e.g., image rotation, flipping, translation, and so on). Despite the effectiveness of data augmentation, few approaches have been explored in graph domain due to its non-Euclidean nature and dependencies between data samples. Simple approaches exist such as DropEdge [23] to randomly remove a certain number of edges and AdaEdge [16] to adaptively control the inter-class/intra-class edges. Similarly, a method to propagate the perturbed node features by randomly dropping on a node-based was proposed in [24]. GAug [17] proposes the neural edge predictors as an augmentation module. Unlike existing methods employing simple perturbation [23] or extra augmentor model [17, 25], we propose the sampling-based augmentation, where a sequence of augmented samples are drawn from the *explicitly* designed target distribution for augmentation.

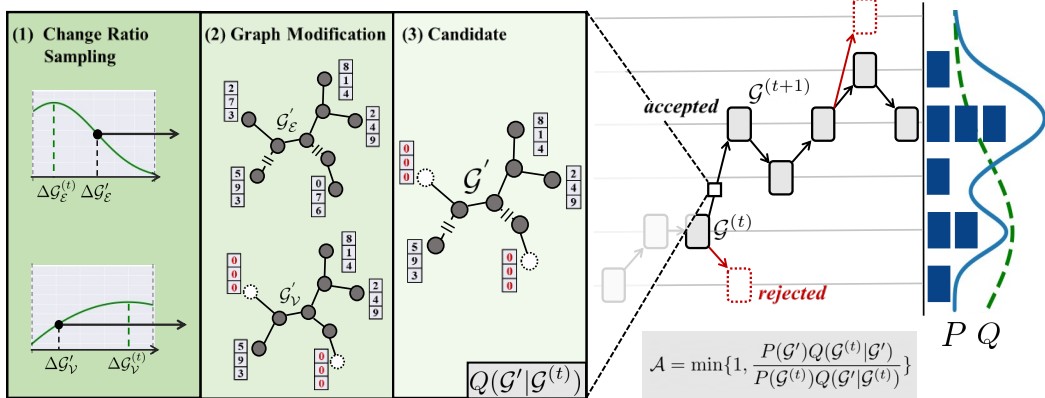

Figure 1: **Sampling process of MH-Aug.** MH-Aug produces augmented samples in two steps. First, it draws a candidate graph $\mathcal{G}'$ from a proposal distribution $Q$ (green). Then, it decides whether to accept or reject the candidate by the acceptance ratio $\mathcal{A}$ calculated by $P$ (blue) and $Q$. The left box shows the details of sampling a candidate graph $\mathcal{G}'$ from the proposal distribution $Q(\mathcal{G}'|\mathcal{G}^{(t)})$ given a current sample $\mathcal{G}^{(t)}$: (1) **Change Ratio Sampling** draws the change ratios $\Delta\mathcal{G}'_{\mathcal{E}}$ and $\Delta\mathcal{G}'_{\mathcal{V}}$ of a candidate graph w.r.t. edges and nodes from the Gaussian distributions truncated to the range $[0, 1]$, (2) **Graph Modification** generates $\mathcal{G}'_{\mathcal{E}}$ and $\mathcal{G}'_{\mathcal{V}}$ by applying the change ratio to the original graph $\mathcal{G}$, and (3) **Candidate** augmented graph $\mathcal{G}'$ is constructed by merging two augmented graphs $\mathcal{G}'_{\mathcal{E}}$ and $\mathcal{G}'_{\mathcal{V}}$.

## 3 Method

We present a novel data augmentation framework for graph-structured data via Metropolis-Hastings algorithm. MH-Aug is a sampling-based augmentation, where a sequence of augmented samples are drawn from the explicit target distribution that enables flexible control of strength and diversity of augmentation. The overall sampling process of MH-Aug is described in Figure 1. In this section, we first summarize the basics for our framework and delineate the components of MH-Aug. Then, we outline the training procedure with proposed consistency regularizers for semi-supervised learning. Lastly, we theoretically prove the distribution of augmented samples by MH-Aug converges to the desired target distribution.

### 3.1 Preliminaries

**Ego-graph, $\mathcal{G}_i$.** A graph is denoted as $\mathcal{G} = (\mathcal{V}, \mathcal{E})$, where $\mathcal{V}$ and $\mathcal{E}$ are the sets of nodes and edges. A $k$-hop ego-graph $\mathcal{G}_i$ [26] is a subgraph of $\mathcal{G}$ centered at node $v_i \in \mathcal{V}$, consisting of neighbors within $k$ hops from node $v_i$ and all edges between the neighbors including $v_i$. In other words, the $k$-hop ego-graph of a node $v_i$ is defined as $\mathcal{G}_i = (\mathcal{V}_i, \mathcal{E}_i)$, $\mathcal{V}_i = \{u | S(u, v_i) \le k, u \in \mathcal{V}\}$, $\mathcal{E}_i = \{(u, v) | (u, v) \in \mathcal{E}$ and $u, v \in \mathcal{V}_i\}$, where $S(u, v)$ is the length of the shortest path between nodes $u$ and $v$. In this paper, we do not explicitly specify $k$ for ego-graphs since 2-hop ego-graphs are used in all experiments.

**Change ratio of graph, $\Delta\mathcal{G}'$.** The change ratio of graph $\mathcal{G}$ to $\mathcal{G}' = (\mathcal{V}', \mathcal{E}')$ is measured by the number of added/deleted edges (or nodes) divided by the number of original edges (or nodes), *i.e.*, $\Delta\mathcal{G}'_{\mathcal{E}} = (|\mathcal{E}' - \mathcal{E}| + |\mathcal{E} - \mathcal{E}'|)/|\mathcal{E}|$ and $\Delta\mathcal{G}'_{\mathcal{V}} = (|\mathcal{V}' - \mathcal{V}| + |\mathcal{V} - \mathcal{V}'|)/|\mathcal{V}|$. Since, in this work, we consider only subgraphs of the original input graph as augmented samples, which is similar to DropEdge [23] and DropNode [24], the change ratio can be equivalently written as $\Delta\mathcal{G}'_{\mathcal{E}} = 1 - |\mathcal{E}'|/|\mathcal{E}|$ and $\Delta\mathcal{G}'_{\mathcal{V}} = 1 - |\mathcal{V}'|/|\mathcal{V}|$. Thereby $\Delta\mathcal{G}'_{\mathcal{E}}$ and $\Delta\mathcal{G}'_{\mathcal{V}}$ are always ranged in $[0, 1]$.

**Metropolis-Hastings (MH) algorithm.** MH algorithm is a Markov chain Monte Carlo method to draw random samples from a target distribution when direct sampling is difficult [27]. The algorithm comprises three components: the target distribution $P$, the proposal distribution $Q$, and the acceptance ratio $\mathcal{A}$. The MH algorithm iteratively draws samples from the target distribution $P$ being only dependent on the current sample. The MH algorithm uses a proposal distribution $Q$ to draw a candidate sample and evaluates the acceptance ratio $\mathcal{A}$ to decide whether to accept or reject the candidate sample. The accepted samples by the MH algorithm follow the target distribution $P$.

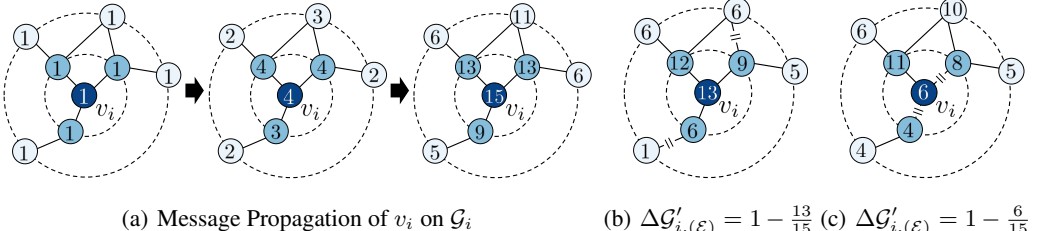

(a) Message Propagation of $v_i$ on $\mathcal{G}_i$      (b) $\Delta\mathcal{G}'_{i,(\mathcal{E})} = 1 - \frac{13}{15}$ (c) $\Delta\mathcal{G}'_{i,(\mathcal{E})} = 1 - \frac{6}{15}$

Figure 2: **Calculation of $\Delta\mathcal{G}'_{i,(\mathcal{E})}$.** (a) displays the message propagation of $v_i$ on ego-graph $\mathcal{G}_i$ in order. The received message of $v_i$ is 15 in the original 2-hop $\mathcal{G}_i$. In case of dropping distant (2-hop) edges (b), the received message of $v_i$ is 13 and $\Delta\mathcal{G}'_{i,(\mathcal{E})}$ becomes 0.13. In case of dropping same number of near (1-hop) edges (c), the received message of $v_i$ is 6 and $\Delta\mathcal{G}'_{i,(\mathcal{E})}$ becomes 0.6.

## 3.2 Metropolis-Hastings Data Augmentation

Our objective is to sample the augmented graph $\mathcal{G}'$ from the target distribution $P$ given the original graph $\mathcal{G}$ and can be written as

$$\mathcal{G}' \sim P(\mathcal{G}'; \mathcal{G}). \tag{1}$$

Since direct sampling from the target distribution $P$ is challenging, we propose a novel data augmentation method based on Metropolis-Hastings algorithm.

**Target Distribution.** We design the target distribution $P$ to control the strength and diversity of augmentation for effective learning. The strength and diversity can be discussed from two perspectives: a full graph and ego-graphs. In our framework, the strength of augmentation is measured by the change ratio of ego-graphs, *i.e.*, $\Delta\mathcal{G}'_i$, since most existing GNNs with $k$-layers learn node representations based on their $k$-hop ego-graphs. On the other hand, the diversity of augmentation is controlled by $\Delta\mathcal{G}'$ and $\Delta\mathcal{G}'_i$ from both full graph and ego-graph perspectives. The diversity of augmentation is adaptively adjusted for each ego-graph by the standard deviation $\sigma(\epsilon_i)$ that is a simple linear function of the entropy $\epsilon_i$ of the prediction at node $v_i$. Given the expected strength $\mu_\mathcal{E} \in \mathbb{R}$ and ego-graph level diversity $\sigma_\mathcal{E}(\epsilon_i) \in \mathbb{R}$, the target distribution $P$ w.r.t. edges is given as follows:

$$P_\mathcal{E}(\mathcal{G}') \propto \left[ \prod_i^{|\mathcal{V}|} \exp\left( -\frac{(\Delta\mathcal{G}'_{i,(\mathcal{E})} - \mu_\mathcal{E})^2}{2\{\sigma_\mathcal{E}(\epsilon_i)\}^2} \right) \right]^{\lambda_1} \cdot \left[ \frac{1}{\binom{|\mathcal{E}|}{|\mathcal{E}| \cdot \Delta\mathcal{G}'_\mathcal{E}}} \right]^{\lambda_2}, \tag{2}$$

where $\Delta\mathcal{G}'_{i,(\mathcal{E})}$ is the change ratio of $\mathcal{G}'_i$ w.r.t. $\mathcal{E}$ and $\lambda$s are hyperparamters for controlling the influence of the two components. To have various full graph change ratios $\Delta\mathcal{G}'_\mathcal{E}$, the normalization by the number of possible augmented graphs corresponding to the same change ratio, $\binom{|\mathcal{E}|}{|\mathcal{E}| \cdot \Delta\mathcal{G}'_\mathcal{E}}$, is necessary. As the size of graph increases, without the normalization, it becomes extremely difficult to generate augmented samples with a low (or high) full graph change ratio. For more details, see Section 4.2. Similarly, The target distribution $P$ with respect to nodes can be written as follows:

$$P_\mathcal{V}(\mathcal{G}') \propto \left[ \prod_i^{|\mathcal{V}|} \exp\left( -\frac{(\Delta\mathcal{G}'_{i,(\mathcal{V})} - \mu_\mathcal{V})^2}{2\{\sigma_\mathcal{V}(\epsilon_i)\}^2} \right) \right]^{\lambda_3} \cdot \left[ \frac{1}{\binom{|\mathcal{V}|}{|\mathcal{V}| \cdot \Delta\mathcal{G}'_\mathcal{V}}} \right]^{\lambda_4}, \tag{3}$$

where $\Delta\mathcal{G}'_{i,(\mathcal{V})}$ is the change ratio of $\mathcal{G}'_i$ w.r.t. the nodes, $\mathcal{V}$. With combining the two distributions, the overall target distribution is defined as:

$$P(\mathcal{G}') = P_\mathcal{E}(\mathcal{G}') \cdot P_\mathcal{V}(\mathcal{G}'). \tag{4}$$

In our experiment, unlike the change ratio of the full graph $\Delta\mathcal{G}'_\mathcal{E}$ (and $\Delta\mathcal{G}'_\mathcal{V}$), we define the *ego-graph change ratio* $\Delta\mathcal{G}'_{i,(\mathcal{E})}$ (and $\Delta\mathcal{G}'_{i,(\mathcal{V})}$) with the change of the number of received messages from $k$-hop ego-graphs. Figure 2 illustrates the calculation of $\Delta\mathcal{G}'_{i,(\mathcal{E})}$ regarding two different cases: (b) dropping distant (2-hop) edges and (c) dropping near (1-hop) edges. In this definition, even if the number of dropped edges is the same, dropping edges connecting nodes closer to the center node $v_i$ leads to a larger $\Delta\mathcal{G}'_{i,(\mathcal{E})}$ than the case of distant nodes ($0.6 > 0.13$), which can be regarded as a much stronger

augmentation. It indicates that the amount of received messages depends on not only the number of removed edges but also which edges are dropped. This *structural property* can only be properly handled from ego-graph perspective. In practice, this definition allows time and memory efficient implementation using matrix multiplications as

$$\Delta \mathcal{G}'_{i,(\mathcal{E})} = 1 - \frac{(\tilde{A}'^k \mathbb{1})_i}{(\tilde{A}^k \mathbb{1})_i}, \text{ and } \Delta \mathcal{G}'_{i,(\mathcal{V})} = 1 - \frac{(\tilde{A}^k \mathbf{m})_i}{(\tilde{A}^k \mathbb{1})_i}, \tag{5}$$

where $\tilde{A}$ and $\tilde{A}'$ are adjacency matrices of the original graph $\mathcal{G}$ and the current graph $\mathcal{G}'$, where both graphs include a self-connection for every node, $\mathbb{1} \in \mathbb{R}^{|\mathcal{V}|}$ is a vector of ones, and $\mathbf{m} \in \mathbb{R}^{|\mathcal{V}|}$ is a mask vector for DropNode.

**Proposal Distribution.** For efficient sampling and a theoretical guarantee of convergence to the target distribution, a proposal distribution is crucial. A proposal distribution $Q(\mathcal{G}'|\mathcal{G}^{(t)})$ suggests a candidate augmented sample $\mathcal{G}'$, given the current sample $\mathcal{G}^{(t)}$. To draw diversely augmented graphs with various edge/node change ratios $\Delta \mathcal{G}'_{\mathcal{E}}$ and $\Delta \mathcal{G}'_{\mathcal{V}}$, a candidate augmented sample is generated by three steps: 1) change ratio sampling, 2) graph modification and 3) merging. We first independently *sample change ratios* $\Delta \mathcal{G}'_{(\cdot)}$ for edges and nodes from Gaussian distributions truncated to the range $[0, 1]$ given mean $\Delta \mathcal{G}^{(t)}_{(\cdot)}$ and standard deviation $\sigma_{\Delta,(\cdot)}$. Then, we *modify the original graph* $\mathcal{G}$ to generate augmented samples $\mathcal{G}'_{\mathcal{E}}$ and $\mathcal{G}'_{\mathcal{V}}$, which can be viewed as a uniform sampling from all possible augmented graphs with $\Delta \mathcal{G}'_{\mathcal{E}}$ and $\Delta \mathcal{G}'_{\mathcal{V}}$, respectively. Finally, the two graphs $\mathcal{G}'_{\mathcal{E}}$ and $\mathcal{G}'_{\mathcal{V}}$ are *merged* to construct the candidate augmented sample $\mathcal{G}'$. Formally, the proposal distribution is given by:

$$Q(\mathcal{G}'|\mathcal{G}^{(t)}) \propto \frac{\phi(\xi'_{\mathcal{E}})}{\Phi(\beta_{\mathcal{E}}) - \Phi(\alpha_{\mathcal{E}})} \cdot \frac{\phi(\xi'_{\mathcal{V}})}{\Phi(\beta_{\mathcal{V}}) - \Phi(\alpha_{\mathcal{V}})} \cdot \frac{1}{\binom{|\mathcal{E}|}{|\mathcal{E}| \cdot \Delta \mathcal{G}'_{\mathcal{E}}}} \cdot \frac{1}{\binom{|\mathcal{V}|}{|\mathcal{V}| \cdot \Delta \mathcal{G}'_{\mathcal{V}}}}, \tag{6}$$

where $\alpha_{(\cdot)} = \frac{a - \Delta \mathcal{G}^{(t)}_{(\cdot)}}{\sigma_\Delta}$, $\beta_{(\cdot)} = \frac{b - \Delta \mathcal{G}^{(t)}_{(\cdot)}}{\sigma_\Delta}$, $\xi'_{(\cdot)} = \frac{\Delta \mathcal{G}'_{(\cdot)} - \Delta \mathcal{G}^{(t)}_{(\cdot)}}{\sigma_\Delta}$, $\phi(x) = \frac{1}{\sqrt{2\pi}} \exp(-\frac{1}{2}x^2)$ as the probability density function of the standard normal distribution and $\Phi(x) = \frac{1}{2}(1 + erf(\frac{x}{\sqrt{2}}))$ as its cumulative distribution function. $a$ and $b$ represent the extremes of Gaussian distribution. Since the change ratio should be ranged in $[0, 1]$, $a$ is 0 and $b$ is 1. In (6), the first and second terms denote the likelihood of the change ratios $\Delta \mathcal{G}'_{\mathcal{E}}$ and $\Delta \mathcal{G}'_{\mathcal{V}}$ given $\Delta \mathcal{G}^{(t)}$. The third and fourth terms are for the probability of a sample with $\Delta \mathcal{G}'_{\mathcal{E}}$ and $\Delta \mathcal{G}'_{\mathcal{V}}$.

**Acceptance Ratio.** Starting with the original graph $\mathcal{G}$, MH-Aug draws the candidate graph $\mathcal{G}'$ from the proposal distribution $Q$. Then, with an acceptance ratio $\mathcal{A}$, MH-Aug decides whether to accept or reject the candidate $\mathcal{G}'$. $\mathcal{A}$ is given as:

$$\mathcal{A} = \min \left\{ 1, \frac{P(\mathcal{G}')Q(\mathcal{G}^{(t)}|\mathcal{G}')}{P(\mathcal{G}^{(t)})Q(\mathcal{G}'|\mathcal{G}^{(t)})} \right\}. \tag{7}$$

The computation of $\mathcal{A}$ with target distribution in (4) and proposal distribution in (6) is described in the supplement. MH-Aug generates a sequence of augmented graphs $\{\mathcal{G}^{(t)}\}_{0 \le t \le T}$, where $T$ is the number of accepted samples.

### 3.3 Consistency Training with MH-Aug

Inspired by recent works [19, 28, 20] that show the importance of advanced augmentation methods in leveraging unlabeled data, we demonstrate the effectiveness of our augmentation method in both supervised and semi-supervised settings. Similar to consistency regularization [20], we propose a simple training strategy with the following regularizers:

$$\mathcal{L}_u = \frac{1}{|\mathcal{V}|} \sum_i^{|\mathcal{V}|} D_{KL} \left[ f(\mathcal{G}_i^{(t)}; \theta) \, || \, f(\mathcal{G}_i^{(t+1)}; \theta) \right], \text{ and } \mathcal{L}_h = \frac{1}{|\mathcal{V}|} \sum_i^{|\mathcal{V}|} [-f(\mathcal{G}_i; \theta) \log(f(\mathcal{G}_i; \theta))], \tag{8}$$

where $D_{KL}(\cdot \, || \, \cdot)$ is the Kullback–Leibler divergence, $f(\cdot)$ is the GNNs parameterized by $\theta$ and $\mathcal{G}_i$ is the $k$-hop ego-graph for node $i$. $\mathcal{L}_u$ encourages the consistency of predictions on two consecutive

augmented samples $\mathcal{G}_i^{(t)}$, and $\mathcal{G}_i^{(t+1)}$. $\mathcal{L}_h$ penalizes unconfident predictions and sharpens predictions. The two regularizers can be applied to both labeled and unlabeled nodes in the node classification task. With the two regularizers and the standard cross-entropy loss $\mathcal{L}_s$ for supervised samples, the overall loss for semi-supervised learning is given as

$$\mathcal{L} = \mathcal{L}_s + \gamma_1 \mathcal{L}_u + \gamma_2 \mathcal{L}_h. \tag{9}$$

Our framework is outlined in Algorithm 1. Starting from original graph $\mathcal{G}$ with 0 change $\Delta\mathcal{G}'$, MH-Aug generates new augmented graph data $\mathcal{G}'$ with the change of $\Delta\mathcal{G}'$. It decides whether to accept or reject the candidate $\mathcal{G}'$ with acceptance score $\mathcal{A}$. GNN models are trained with the accepted augmented data with our loss in (9). Then, the process is repeated until the model converges.

---

**Algorithm 1** Metropolis-Hastings Data Augmentation (MH-Aug) Framework

---

**Input:** target distribution $P$, proposal distribution $Q$, original graph $\mathcal{G}$
**Output:** network parameter $\theta$

> **Initialize** $t \leftarrow 0, \mathcal{G}^{(0)} \leftarrow \mathcal{G}$
> **while** not convergence **do**
>> Draw $\mathcal{G}'$ from $Q(\mathcal{G}'|\mathcal{G}^{(t)})$            ▷ Q in Eq.(6)
>> Draw $u$ from $Uniform(0,1)$
>> **if** $u \leq \mathcal{A}$ **then**            ▷ $\mathcal{A}$ in Eq.(7)
>>> $\mathcal{G}^{(t+1)} \leftarrow \mathcal{G}'$
>>> Update $\theta$ with $\mathcal{L}(\mathcal{G}, \mathcal{G}^{(t)}, \mathcal{G}^{(t+1)}; \theta)$            ▷ $\mathcal{L}$ in Eq.(9)
>>> $t \leftarrow t + 1$
>> **end if**
> **end while**

---

### 3.4 Theoretical Analysis

The goal of the Metropolis-Hastings algorithm is to generate a sequence of samples according to a desired target distribution $P$. To accomplish this, the Metropolis-Hastings algorithm uses a Markov process, which asymptotically reaches a unique stationary distribution $\pi(x)$ such that $\pi(x) = P(x)$ [29]. Here, we show that Markov chain of MH-Aug, which has a sequence of augmented graph as states, converges to the unique and stationary target distribution $P(\mathcal{G}')$ defined in (4).

**Lemma 3.1.** *Let the sequence of augmented graphs $\{\mathcal{G}^{(t)}\}_{0 \leq t \leq T}$ be the Markov chain produced by MH-Aug. If we define the acceptance ratio $\mathcal{A}$ with target distribution $P$ in* (4) *and proposal distribution $Q$ in* (6)*, the sequence converges to a unique stationary target distribution $P$.*

This can be drawn from the Convergence theorem of Markov chain [30]. The proof is in the supplement. By Lemma 3.1, we theoretically show augmented samples of MH-Aug converges to our desired target distribution. Our toy examples show a sequence of augmented graphs actually converges well to the target distribution (see Section 4.2 for details).

## 4 Experiments

In this section, we demonstrate the effectiveness of MH-Aug on various benchmark datasets. We start with describing datasets, baselines, and implementation details for the experiments. Next, we evaluate our framework for node classification in Section 4.1 and we offer qualitative analyses in Section 4.2 on three parts: effectiveness of ego-graph perspective for desired target distribution $P$, necessity of normalization term in $P$, and whether generated samples from MH-Aug converge to $P$.

**Datasets.** We evaluate our method on five benchmark datasets in three categories: (1) Citation networks: CORA and CITESEER [31], (2) Amazon product networks: Computers and Photo [32], and (3) Coauthor Networks: CS [32]. We follow the standard data split protocol in the transductive settings for node classification, *e.g.*, [4] for CORA and CITESEER and [32] for the rest.

**Baselines.** As backbone models to validate MH-Aug, we adopt three standard graph neural networks: GCN [4], GraphSAGE [1], and GAT [3]. We compare our method with vanilla models (without

Table 1: Node classification results. Mean accuracy and standard deviation from 10 repetitions are reported. We compare our methods with baselines of two categories: 1) supervised learning with augmentation (*e.g.*, DropEdge and AdaEdge), which are comparable to our MH-Aug (w/o Reg) and 2) semi-supervised learning (*e.g.*, GAug, SSL, BVAT, UDA* and GraphMix) that are comparable to our MH-Aug (w/ Reg). For each dataset and baseGNN the highest score is marked in **bold**.

| BaseGNNs | Method | DATASET | | | | |
| --- | --- | --- | --- | --- | --- | --- |
| | | **CORA** | **CITESEER** | **Compu.** | **Photo** | **CS** |
| GCN | Vanilla | $81.54_{\pm0.76}$ | $71.64_{\pm0.31}$ | $79.68_{\pm2.16}$ | $89.02_{\pm1.49}$ | $91.45_{\pm0.28}$ |
| | DropEdge [23] | $82.21_{\pm0.71}$ | $71.93_{\pm0.31}$ | $80.59_{\pm1.75}$ | $89.33_{\pm1.58}$ | $91.69_{\pm0.43}$ |
| | AdaEdge [16] | $82.30_{\pm0.80}^{\dagger}$ | $69.70_{\pm0.90}^{\dagger}$ | $80.66_{\pm1.22}$ | $\mathbf{89.94_{\pm0.84}}$ | $90.30_{\pm0.40}^{\dagger}$ |
| | MH-Aug (w/o Reg) | $\mathbf{83.55_{\pm0.34}}$ | $\mathbf{72.96_{\pm0.48}}$ | $\mathbf{80.95_{\pm2.03}}$ | $89.65_{\pm1.67}$ | $\mathbf{91.81_{\pm0.33}}$ |
| | GAug-M [17] | $83.50_{\pm0.40}^{\dagger}$ | $72.30_{\pm0.40}^{\dagger}$ | $78.90_{\pm1.76}$ | $88.46_{\pm1.24}$ | OOM |
| | GAug-O [17] | $83.60_{\pm0.50}^{\dagger}$ | $73.30_{\pm1.10}^{\dagger}$ | OOM | $89.04_{\pm1.18}$ | OOM |
| | SSL [21] | $83.80_{\pm0.73}^{\dagger}$ | $72.95_{\pm0.62}^{\dagger}$ | - | - | - |
| | BVAT [22] | $83.60_{\pm0.50}^{\dagger}$ | $74.00_{\pm0.60}^{\dagger}$ | $80.07_{\pm2.41}$ | $88.46_{\pm2.25}$ | $92.21_{\pm0.37}$ |
| | UDA*[19] | $83.59_{\pm0.61}$ | $73.56_{\pm0.41}$ | $81.68_{\pm2.95}$ | $89.95_{\pm1.73}$ | $92.26_{\pm0.37}$ |
| | GraphMix [18] | $83.90_{\pm0.57}^{\dagger}$ | $74.70_{\pm0.59}^{\dagger}$ | $80.72_{\pm1.16}$ | $89.05_{\pm1.01}$ | $91.83_{\pm0.51}^{\dagger}$ |
| | MH-Aug (w/ Reg) | $\mathbf{85.16_{\pm0.35}}$ | $\mathbf{75.49_{\pm0.29}}$ | $\mathbf{82.80_{\pm2.08}}$ | $\mathbf{90.87_{\pm1.49}}$ | $\mathbf{92.60_{\pm0.43}}$ |
| GraphSAGE | Vanilla | $79.78_{\pm0.74}$ | $71.09_{\pm0.59}$ | $79.59_{\pm1.84}$ | $89.10_{\pm1.60}$ | $91.35_{\pm1.00}$ |
| | DropEdge [23] | $80.36_{\pm0.80}$ | $71.46_{\pm0.57}$ | $79.87_{\pm1.87}$ | $89.86_{\pm1.78}$ | $91.84_{\pm0.76}$ |
| | AdaEdge [16] | $80.20_{\pm1.20}^{\dagger}$ | $69.40_{\pm0.80}^{\dagger}$ | $80.43_{\pm1.30}$ | $\mathbf{90.57_{\pm0.70}}$ | $90.30_{\pm0.40}^{\dagger}$ |
| | MH-Aug (w/o Reg) | $\mathbf{82.61_{\pm0.66}}$ | $\mathbf{72.12_{\pm0.99}}$ | $\mathbf{81.74_{\pm2.52}}$ | $90.37_{\pm1.50}$ | $\mathbf{92.27_{\pm0.49}}$ |
| | GAug-M [17] | $83.20_{\pm0.40}^{\dagger}$ | $71.20_{\pm0.40}^{\dagger}$ | $79.84_{\pm1.99}$ | $88.72_{\pm0.97}$ | OOM |
| | GAug-O [17] | $82.00_{\pm0.50}^{\dagger}$ | $72.70_{\pm0.70}^{\dagger}$ | OOM | $88.16_{\pm2.70}$ | OOM |
| | BVAT [22] | $83.12_{\pm0.64}$ | $72.23_{\pm0.46}$ | $78.72_{\pm2.73}$ | $89.40_{\pm1.79}$ | $92.63_{\pm0.48}$ |
| | UDA*[19] | $83.37_{\pm0.29}$ | $75.16_{\pm0.16}$ | $82.16_{\pm2.00}$ | $90.61_{\pm2.00}$ | $92.83_{\pm0.39}$ |
| | GraphMix [18] | $82.28_{\pm0.55}$ | $69.62_{\pm0.36}$ | $81.33_{\pm1.46}$ | $88.46_{\pm1.36}$ | $89.29_{\pm0.45}$ |
| | MH-Aug (w/ Reg) | $\mathbf{84.70_{\pm0.39}}$ | $\mathbf{75.55_{\pm0.44}}$ | $\mathbf{83.62_{\pm2.60}}$ | $\mathbf{92.19_{\pm1.37}}$ | $\mathbf{93.61_{\pm0.58}}$ |
| GAT | Vanilla | $82.23_{\pm0.46}$ | $71.37_{\pm0.93}$ | $78.47_{\pm1.86}$ | $87.80_{\pm1.36}$ | $90.90_{\pm0.31}$ |
| | DropEdge [23] | $83.04_{\pm0.37}$ | $72.16_{\pm0.91}$ | $81.04_{\pm1.86}$ | $88.73_{\pm1.54}$ | $91.10_{\pm0.37}$ |
| | AdaEdge [16] | $77.90_{\pm2.00}^{\dagger}$ | $69.10_{\pm0.80}^{\dagger}$ | $77.52_{\pm1.72}$ | $88.92_{\pm0.87}$ | $86.60_{\pm0.16}^{\dagger}$ |
| | MH-Aug (w/o Reg) | $\mathbf{83.49_{\pm0.69}}$ | $\mathbf{72.81_{\pm0.98}}$ | $\mathbf{81.72_{\pm1.66}}$ | $\mathbf{90.23_{\pm0.97}}$ | $\mathbf{91.40_{\pm0.39}}$ |
| | GAug-M [17] | $82.10_{\pm1.00}^{\dagger}$ | $71.50_{\pm0.50}^{\dagger}$ | $77.70_{\pm2.10}$ | $87.08_{\pm1.00}$ | OOM |
| | GAug-O [17] | $82.20_{\pm0.80}^{\dagger}$ | $71.60_{\pm1.10}^{\dagger}$ | OOM | $86.45_{\pm1.52}$ | OOM |
| | SSL [21] | $83.70_{\pm0.61}^{\dagger}$ | $72.73_{\pm0.72}^{\dagger}$ | - | - | - |
| | UDA*[19] | $83.71_{\pm0.48}$ | $73.24_{\pm0.48}$ | $82.42_{\pm2.95}$ | $89.79_{\pm1.36}$ | $91.78_{\pm0.23}$ |
| | GraphMix [18] | $83.32_{\pm0.18}^{\dagger}$ | $73.08_{\pm0.23}^{\dagger}$ | - | - | - |
| | MH-Aug (w/ Reg) | $\mathbf{84.95_{\pm0.40}}$ | $\mathbf{75.53_{\pm0.32}}$ | $\mathbf{83.25_{\pm1.88}}$ | $\mathbf{90.61_{\pm1.34}}$ | $\mathbf{92.08_{\pm0.58}}$ |

UDA* denotes our extension of UDA in the graph domain. † denotes the results reported in the original paper.

augmentation), augmentation-based supervised learning (DropEdge [23], AdaEdge [16]), and semi-supervised learning framework (GAug [17], SSL [21], BVAT [22], UDA* [19], GraphMix [18]). In the case of DropEdge [23] and AdaEdge [16], they use only cross-entropy loss (supervised setting) while the rest of models employs extra loss functions for regularization (semi-supervised setting).

## 4.1 Results on Node Classification

Table 1 shows the experimental results on node classification with five datasets compared to baseline models. We implemented all the baselines and conducted experiments for fair comparison except for the case where the performance (marked with †) is available in the original papers [16, 17, 21, 22, 18]. Also, we denote out-of-memory as OOM. MH-Aug (w/o Reg) means training the model only with the labeled data and cross-entropy loss $\mathcal{L}_s$ whereas MH-Aug (w/ Reg) means using extra regularization losses to explicitly utilize the unlabeled data. Our full framework MH-Aug (w/ Reg), which is trained in the semi-supervised setting, consistently achieves the best performance in all datasets and the

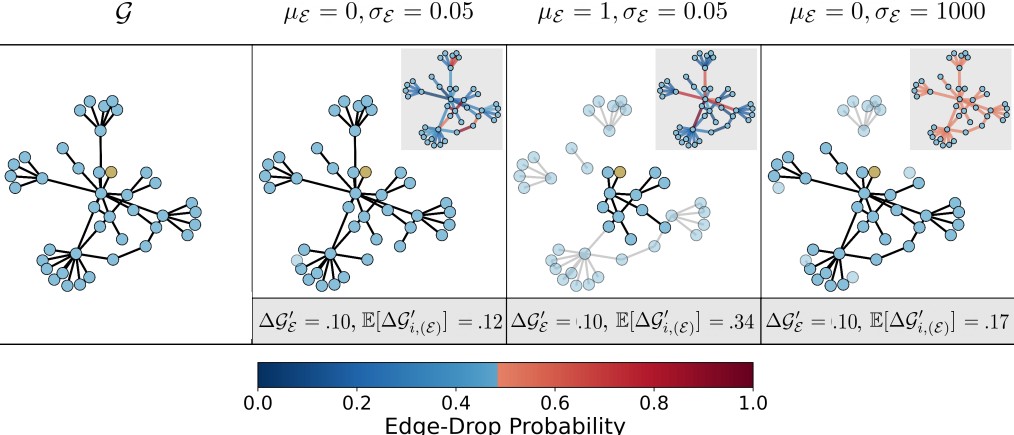

Figure 3: **Diverse $\mathcal{G}'$ sampled by MH-Aug**. The first cell is the original graph $\mathcal{G}$ extracted from CORA. With the fixed full graph change ratio $\Delta\mathcal{G}'_{\mathcal{E}}$, augmented graphs with different $\mu_{\mathcal{E}}$ and $\sigma_{\mathcal{E}}$ are generated by MH-Aug. All graphs above are 3-hop ego-graphs with a center node marked as yellow. Nodes and edges which are not in the ego-graph after augmentation are blurred. Mini-maps at the upper right corner is the edge-drop probability, where blue means higher probability and red means low probability to drop the edge. By explicitly controlling the strength and diversity from an ego-graph perspective, MH-Aug generates diverse augmentations.

improvement against the vanilla models is 3.16% on average. In particular, we observe that MH-Aug improves the performance by 4.92% compared to the vanilla GraphSAGE on CORA. In addition, MH-Aug provides an 4.16% gain on CITESEER on average over all models (*i.e.*, vanilla GCN, GraphSAGE and GAT).As an ablation study, we conduct experiments with MH-Aug (w/o Reg), our framework trained in the supervised setting. Table 1 shows that MH-Aug (w/o Reg) achieves 1.47% improvement on average compared to vanilla models. MH-Aug (w/o Reg) provides considerable gain over all dataset and model. More specifically, it provides 3.25% performance improvement compared to the vanilla GAT model on Computers. In addition, MH-Aug (w/o Reg) beats DropEdge for all settings and mostly beats AdaEdge that optimizes the graph topology based on the model predictions. It is worth noting that even though MH-Aug (w/o Reg) does not explicitly utilize unlabeled data during training, MH-Aug (w/o Reg) achieves competitive performance compared to other semi-supervised methods, especially in the following cases: GCN on CORA (83.55%); GraphSAGE on CITESEER (72.12%); and GAT on CORA (83.49%), and Photo (90.23%). This demonstrates the effectiveness of our sampling-based augmentation. More discussion on ablation study is in the supplement.

### 4.2 Analysis

**Effectiveness of Ego-graph Perspective.** To validate the effectiveness of ego-graph perspective augmentation, we qualitatively analyze augmented samples by MH-Aug on real data with various settings as shown in Figure 3. An original sample (first column) is a 3-hop ego-graph from CORA. Augmented samples are generated from $\mathcal{G}$ in three settings: $(\mu_{\mathcal{E}}, \sigma_{\mathcal{E}}) = (0,0.05)$, $(\mu_{\mathcal{E}}, \sigma_{\mathcal{E}}) = (1,0.05)$ and $(\mu_{\mathcal{E}}, \sigma_{\mathcal{E}}) = (0,1000)$. The mini maps at the upper right corner shows the edge-drop probability, calculated from $\prod_i^{|\mathcal{V}|} \exp\left(-\frac{(\Delta\mathcal{G}'_{i,(\mathcal{E})} - \mu_{\mathcal{E}})^2}{2\{\sigma_{\mathcal{E}}\}^2}\right)$ of (2). To evaluate the effect of $\mu_{\mathcal{E}}$ and $\sigma_{\mathcal{E}}$ w.r.t. ego-graph, we fix the full-graph change ratio $\Delta\mathcal{G}'_{\mathcal{E}}$ and observe the expected value of $\Delta\mathcal{G}'_{i,(\mathcal{E})}$ over all possible nodes in the ego-graph, $\mathbb{E}[\Delta\mathcal{G}'_{i,(\mathcal{E})}]$. Thus, the number of dropped edges is identical for all the augmented graphs in Figure 3. It demonstrates that even if the number of dropped edges is the same, one can generate diverse samples by controlling $\mu_{\mathcal{E}}$ and $\sigma_{\mathcal{E}}$. When $\mu_{\mathcal{E}}$, which controls the expected augmentation strength, is large, *e.g.*, $\mu_{\mathcal{E}} = 1, \sigma_{\mathcal{E}} = 0.05$, more important edges (*e.g.*, edges acting as bridges between hub nodes) tend to be dropped. This observation exactly matches to our design in Section 3.2, which considers dropping edges near to the center as a strong augmentation. In addition, the mini map in the fourth cell of Figure 3 indicates if $\sigma_{\mathcal{E}}$ increases, the edge-drop probability of the all edges becomes uniform, *i.e.* MH-Aug subsumes DropEdge as a special case. In sum, the ego-graph perspective enables the explicit control of augmentation strength and diversity to make an advanced augmentation.

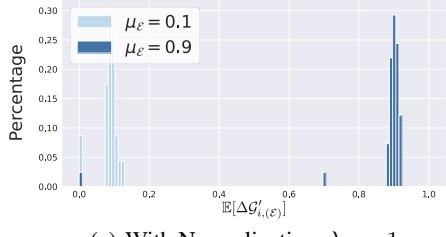
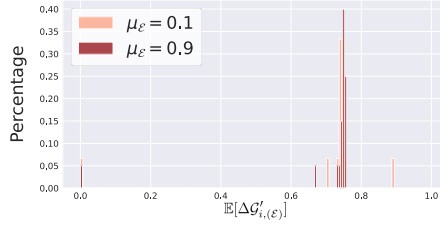

(a) With Normalization, $\lambda_2 = 1$  (b) Without Normalization, $\lambda_2 = 0$

Figure 4: **The effect of normalization term**. The two plots above show the distribution of augmented graphs w.r.t. $\mathbb{E}[\Delta \mathcal{G}'_{i,(\mathcal{E})}]$. We conduct the toy example with the target distribution (a) with normalization (blue) and (b) without normalization (red).

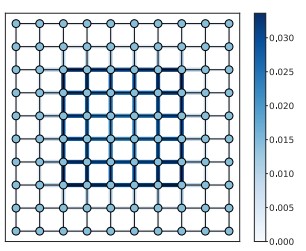
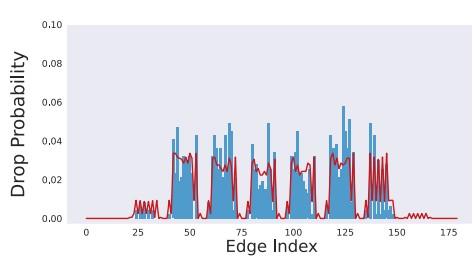

(a) Drop probability of each edges  (b) Visualization of drop probability on grid graphs

Figure 5: **Convergence to the target distribution**. To verify the convergence of MH-Aug, we simulate the sampling procedure of MH-Aug on a grid graph. (a) is the result of target distribution $P$ given $\mu_{\mathcal{E}} = 0$. We highlight the edges according to drop probability. (b) shows that the samples drawn from MH-Aug (blue) follow the target distribution $P$ (red) that we calculate with (2).

**Necessity of Normalization Term.** As mentioned in Section 3.2, the normalization in the target distribution $P$ by the number of possible augmented graphs corresponding to the same change ratio, $\binom{|\mathcal{E}|}{|\mathcal{E}| \cdot \Delta \mathcal{G}'_{\mathcal{E}}}$, is crucial to generating ego-graphs with the desired ego-graph change ratio $\mu_{\mathcal{E}}$ when the number of edges is huge. We demonstrate it with a small but *fully connected* graph to apply MH-Aug. Figure 4 displays the distribution of the empirical mean of $\Delta \mathcal{G}'_{i,(\mathcal{E})}$ from the augmented graph sampled from $P$ with two different $\mu_{\mathcal{E}} = 0.1$ and $\mu_{\mathcal{E}} = 0.9$. With normalization (Figure 4(a)), the sample mean of $\Delta \mathcal{G}'_{i,(\mathcal{E})}$ of both sampling results are near to the $\mu_{\mathcal{E}}$ values. However, without normalization (Figure 4(b)), the empirical mean of $\Delta \mathcal{G}'_{i,(\mathcal{E})}$ remains the same due to overwhelmingly many possible subgraphs with a certain full graph change ratio $\Delta \mathcal{G}'_{\mathcal{E}}$, *e.g.*, $\binom{100 \times 100}{5000}$ for a fully connected graph with 100 nodes. But this does *not* mean that our MH-Aug fails to converges to the target distribution. It merely converges to the undesirable target distribution.

**Convergence to Target Distribution.** In Section. 3.4, we theoretically show that the distribution of samples generated by MH-Aug converges to our desired target distribution $P$. Now, we conduct the experiment to examine whether a sequence of augmented graphs experimentally follows the target distribution. We observe the behavior of MH-Aug with a simple toy example, *i.e.*, a grid graph with 100 nodes. For simplicity, we only consider the change of edges and the target distribution $P_{\mathcal{E}}$ in (2). In 5(b), red line denotes the probability of each edge obtained by calculating $P$ with (2). Blue bars represent the distribution of augmented graphs generated by MH-Aug. It shows MH-Aug generates augmented graphs following the target distribution. In 5(a), we visualize drop probability on graph. Since we set $\mu_{\mathcal{E}}$ small, drop probability of center edges is higher than others.

## 5 Conclusion

We present a novel semi-supervised strategy with Metropolis-Hastings algorithm based augmentation method. This is the first work to impose data augmentation on graph-structured data from a perspective

of a Markov chain Monte Carlo sampling. We theoretically and experimentally show the convergence of augmented samples to target distribution and demonstrate its consistent performance improvement over baselines across five benchmark datasets.

**Acknowledgments.** This work was partly supported by NAVER Corp., National Supercomputing Center with supercomputing resources including technical support (KSC-2021-CRE-0181) and ICT Creative Consilience program (IITP-2021-2020-0-01819) supervised by the IITP.

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
