# Metropolis-Hastings Data Augmentation
# for Graph Neural Networks (Supplement)

**Hyeonjin Park**[1][*]**, Seunghun Lee**[1][*]**, Sihyeon Kim**[1]**, Jinyoung Park**[1]
**Jisu Jeong**[2,3]**, Kyung-Min Kim**[2,3]**, Jung-Woo Ha**[2,3]**, Hyunwoo J. Kim**[1][†]
Korea University[1], NAVER CLOVA[2], NAVER AI LAB[3]
{hyeonjin961030, llsshh319, sh_bs15, lpmn678, hyunwoojkim}@korea.ac.kr
{jisu.jeong, kyungmin.kim.ml, jungwoo.ha}@navercorp.com

## A  Summary

We provide additional experimental results and discussion that are not in the main paper due to the limited space. This supplement includes (1) Reproducibility (*e.g.*, dataset statistics, implementation details and hyperparameter settings), (2) Proof of Lemma 3.1, (3) Additional experiments (*e.g.*, more visualizations about augmented graphs by MH-Aug, further ablation studies and node classification results on large datasets), and (4) Further discussion (*e.g.*, normalization term for our target distribution, calculation of acceptance ratio and negative societal impacts & limitations of our works).

## B  Reproducibility

### B.1  Dataset Statistics

Table 1 summarizes the statistics of five benchmark datasets – Cora[3], Citeseer, Amazon Computers, Amazon Photo, and Coauthor CS and two large datasets for additional experiments – CORA Full and ogbn-arxiv [4]. We use the same experimental settings on every dataset as semi-supervised graph representation learning [1–3].

Table 1: Statistics and details of datasets for node classification.

| Dataset | # Nodes | # Edges | # Features | # Classes | # Train | # Validation | # Test |
|---|---|---|---|---|---|---|---|
| **CORA** | 2,708 | 5,429 | 1,433 | 7 | 140 | 500 | 1,000 |
| **CITESEER** | 3,327 | 4,732 | 3,703 | 6 | 120 | 500 | 1,000 |
| **Amazon Computers** | 13,752 | 245,778 | 767 | 10 | 200 | 300 | 13,252 |
| **Amazon Photo** | 7,650 | 119,043 | 745 | 8 | 160 | 240 | 7,250 |
| **Coauthor CS** | 18,333 | 81,894 | 6,805 | 15 | 300 | 450 | 17,583 |
| **CORA Full** | 19,793 | 65,311 | 8,710 | 70 | 1,311 | 1,800 | 16,682 |
| ogbn-arxiv | 169,343 | 1,166,243 | 128 | 40 | 90,941 | 29,799 | 48,603 |

### B.2  Implementation Details

We conduct 10 trials and report the mean and standard deviation over the trials for every experiment. In details, we set the number of layers to 2 and the dimensionality of hidden representations to 64 for every datasets, except for the dimensionality of GAT hidden representations to 16 and the number of

---

[*]First two authors have equal contribution

[†]is the corresponding author

[3]Licensed under the Creative Commons Attribution 4.0 International (CC BY 4.0) Licence

[4]Copyright (c) 2019 OGB Team. Licensed under MIT License

35th Conference on Neural Information Processing Systems (NeurIPS 2021).

heads to 8. Models are optimized by the Adam [4] optimizer for 2,000 epochs. As for the baselines, we adopt vanilla models, augmentation-based supervised learning (DropEdge[5] [5], AdaEdge [6]), and semi-supervised learning framework (GAug[6] [7], SSL [8], BVAT[7] [9], UDA* [10], GraphMix [11]). We implement the UDA [10] which is a semi-supervised framework for consistency training on image recognition and natural language processing. Herein we directly implement it for graph structured data and denote with a superscript asterisk '*' in the Table 1 from the main paper. For GraphMix with GraphSAGE as base GNNs, we simply adopt the aggregation function of GraphSAGE. The source code for baseline is from the original paper. Our MH-Aug is implemented in Pytorch [12] with the geometric deep learning library Torch-Geometric [13]. All the experiments in this paper are conducted on a single Tesla V100 with 16GB memory size. As for the software version, we use Python 3.8.3, Pytorch 1.5.0 [8] and Pytorch geometric 1.6.2 [9].

## B.3  Hyperparameter Settings

We delineate additional hyperparameters such as coefficients (*i.e.*, $\alpha_{\mathcal{E}}$, $\beta_{\mathcal{E}}$, $\alpha_{\mathcal{V}}$ and $\beta_{\mathcal{V}}$) of the linear function for standard deviation of the target distribution $\sigma_{(\cdot)}(\epsilon_i) = \alpha_{(\cdot)} * \epsilon_i + \beta_{(\cdot)}$ w.r.t. edges $\mathcal{E}$ and nodes $\mathcal{V}$, respective coefficients of regularization for Kullback-Leibler divergence loss $\gamma_1$ and entropy loss $\gamma_2$, the standard deviation of proposal distribution $\sigma_{\Delta,(\cdot)}$ w.r.t. edges and nodes, and the mean of proposal distribution $\mu_{(\cdot)}$ w.r.t. edges and nodes. We empirically perform hyperparameter search for each dataset. Table 2 reports the best hyperparameters of MH-Aug on GCN.

Table 2: Hyperparameters of MH-Aug on GCN.

| Hyperparameters | CORA | CITESEER | Computers | Photo | CS |
|---|---|---|---|---|---|
| $\alpha_{\mathcal{E}}, \beta_{\mathcal{E}}$ | 10, 1 | 5, 1 | 10, 1 | 10, 1 | 0, 0.1 |
| $\alpha_{\mathcal{V}}, \beta_{\mathcal{V}}$ | 1, 1 | 10, 0.1 | 10, 0.1 | 10, 1 | 5, 0.1 |
| $\gamma_1$ | 0.5 | 0.2 | 4 | 3 | 0.2 |
| $\gamma_2$ | 0.5 | 0.5 | 0.4 | 0.5 | 0.4 |
| $\sigma_{\Delta,(\mathcal{E})}, \sigma_{\Delta,(\mathcal{V})}$ | 0.1, 0.01 | 0.005, 0.1 | 0.05, 0.01 | 0.01, 0.005 | 0.001, 0.01 |
| $\mu_{\mathcal{E}}, \mu_{\mathcal{V}}$ | 0.3, 0.7 | 0.2, 0.5 | 0, 0.1 | 0.5, 0 | 0.8, 0.2 |
| $\lambda_1, \lambda_3$ | 5, 1 | 1, 10 | 1, 10 | 1, 10 | 10, 5 |
| $\lambda_2, \lambda_4$ | 0.9999, 0.999 | 0.9999, 0.999 | 0.9999, 0.9999 | 0.9999, 0.9999 | 0.9999, 1 |

## C  Proof of Lemma

**Theorem C.1.** (Convergence Theorem) *Let $X = (X_t)_{t \geq 0}$ be a Markov chain with transition kernel K. If X is **irreducible**, **aperiodic** and **has stationary distribution** $\pi$, then there exist*

$$P(X_t = x | X_0 = x_0) \to \pi(x) \text{ as } t \to \infty, \text{ for all } x \text{ and every initial state } x_0. \quad (1)$$

**Theorem C.2.** (Existence of a Stationary distribution) *A sufficient but not necessary condition is detailed balance, which requires that each transition $x \to x'$ is reversible.*

**Definition C.1.** *A Markov chain with invariant measure $\pi$ is reversible if and only if*

$$\pi_i K_{ij} = \pi_j K_{ji} \quad (2)$$

*for all states i and j, where K is the transition kernel.*

**Lemma 3.1.** *Let the sequence of augmented graphs $\{\mathcal{G}^{(t)}\}_{0 \leq t \leq T}$ be the Markov chain produced by MH-Aug. If we define the acceptance ratio $\mathcal{A}$ with target distribution P in (4) and proposal distribution Q in (6), the sequence converges to a unique stationary target distribution P.*

*Proof of Lemma 3.1.* By Theorem C.1, we can show that MH-Aug converges to the target distribution P, if Markov chain of MH-Aug is *irreducible*, *aperiodic* and *has stationary distribution*. First, we will show the existence of stationary distribution by Theorem C.2 and Definition C.1.

The acceptance ratio $\mathcal{A}(\mathcal{G}, \mathcal{G}')$ of Metropolis-Hastings algorithm can be calculated with the proposal distribution $Q$ and the target distribution $P$. Then, the transition kernel $K(\mathcal{G}'|\mathcal{G})$ can be written as

$$K(\mathcal{G}'|\mathcal{G}) = Q(\mathcal{G}'|\mathcal{G})\mathcal{A}(\mathcal{G}, \mathcal{G}') + \delta_{\mathcal{G}}(\mathcal{G}')r(\mathcal{G}), \tag{3}$$

where $r(\mathcal{G}) = 1 - \int \mathcal{A}(\mathcal{G}, s)Q(s|\mathcal{G})ds$ is the probability of rejecting $\mathcal{G}'$, which is generated by the proposal distribution $Q$ and $\delta_{\mathcal{G}}(\cdot)$ denotes the Dirac delta function located at $\mathcal{G}$. Then, the reversibility can be shown as

$$\begin{aligned}
P(\mathcal{G})K(\mathcal{G}'|\mathcal{G}) &= P(\mathcal{G})Q(\mathcal{G}'|\mathcal{G})\mathcal{A}(\mathcal{G}, \mathcal{G}') \\
&= P(\mathcal{G})Q(\mathcal{G}'|\mathcal{G}) \min\{1, \frac{P(\mathcal{G}')Q(\mathcal{G}|\mathcal{G}')}{P(\mathcal{G})Q(\mathcal{G}'|\mathcal{G})}\} \\
&= \min\{P(\mathcal{G})Q(\mathcal{G}'|\mathcal{G}), P(\mathcal{G}')Q(\mathcal{G}|\mathcal{G}')\},
\end{aligned} \tag{4}$$

$$\begin{aligned}
P(\mathcal{G}')K(\mathcal{G}|\mathcal{G}') &= P(\mathcal{G}')Q(\mathcal{G}|\mathcal{G}')\mathcal{A}(\mathcal{G}', \mathcal{G}) \\
&= P(\mathcal{G}')Q(\mathcal{G}|\mathcal{G}') \min\{1, \frac{P(\mathcal{G})Q(\mathcal{G}'|\mathcal{G})}{P(\mathcal{G}')Q(\mathcal{G}|\mathcal{G}')}\} \\
&= \min\{P(\mathcal{G}')Q(\mathcal{G}|\mathcal{G}'), P(\mathcal{G})Q(\mathcal{G}'|\mathcal{G})\}.
\end{aligned} \tag{5}$$

$$P(\mathcal{G})K(\mathcal{G}'|\mathcal{G}) = \min\{P(\mathcal{G})Q(\mathcal{G}'|\mathcal{G}), P(\mathcal{G}')Q(\mathcal{G}|\mathcal{G}')\} = P(\mathcal{G}')K(\mathcal{G}|\mathcal{G}'). \tag{6}$$

By Definition C.1, $P$ in (4) is reversible and there exists a stationary distribution for Markov chain by Theorem C.2. Note that we do not consider the case of rejection because it is trivial that the detailed balance equation always satisfies when $\mathcal{G} = \mathcal{G}'$. Now, we need to show that the Markov chain of MH-Aug is *irreducible* and *aperiodic*. The transition kernel $K(\mathcal{G}_j|\mathcal{G}_i)$ for all $i, j$ is defined as:

$$K(\mathcal{G}_j|\mathcal{G}_i) = Q(\mathcal{G}_j|\mathcal{G}_i)\mathcal{A}(\mathcal{G}_i, \mathcal{G}_j) + \delta_{\mathcal{G}_i}(\mathcal{G}_j)r(\mathcal{G}_i). \tag{7}$$

Since $Q(\mathcal{G}_j|\mathcal{G}_i)\mathcal{A}(\mathcal{G}_i, \mathcal{G}_j) > 0$ and $\delta_{\mathcal{G}_i}(\mathcal{G}_j)r(\mathcal{G}_i) \geq 0$, Markov chain of MH-Aug has positive probability for all $0 \leq \Delta\mathcal{G}_i \leq 1$ and $0 \leq \Delta\mathcal{G}_j \leq 1$ to reach any range in the support of the proposal distribution. Thus, it is irreducible. Similarly, the probability of $\mathcal{G}_i$ to stay at $\mathcal{G}_i$, which can be written as $K(\mathcal{G}_i|\mathcal{G}_i)$, is strictly greater than 0. Thus, its period is 1, which means that it is aperiodic. Therefore, by Theorem C.1, MH-Aug converges to the stationary target distribution $P$. $\square$

# D   Additional Experiments

## D.1   Further Visualizations of Augmented Graphs by MH-Aug

As shown in Figure 1, we provide more visualizations of augmented graphs by MH-Aug with various settings. Original samples in the first column are a 3-hop ego-graph from CORA and the corresponding augmented samples in the other columns are generated from original samples. Augmented samples are drawn from three different target distributions as same as the main paper. We only consider the edge drop, not the node drop in this visualization for simplicity and fix the number of dropped edges $\Delta\mathcal{G}'_{\mathcal{E}}$ to evaluate the effect of $\mu_{\mathcal{E}}$ and $\sigma_{\mathcal{E}}$ in the target distribution. Mini maps represent the Edge-Drop probability directly calculated from $P$.

In Figure 1, we can verify two things. First, MH-Aug samples the augmented graphs following the target distribution. If the Edge-Drop probability of a specific edge is high according to the target distribution (red), then the corresponding edge tends to be dropped in the augmented graph drawn by MH-Aug. Second, MH-Aug controls the strength and diversity of augmentation with $\mu_{\mathcal{E}}$ and $\sigma_{\mathcal{E}}$. When $\mu_{\mathcal{E}}$, which controls the expected augmentation strength, is large, *e.g.*, $\mu_{\mathcal{E}} = 1, \sigma_{\mathcal{E}} = 0.05$, more important edges (*e.g.*, edges acting as bridges between hub nodes) tend to be dropped. Conversely, when $\mu_{\mathcal{E}}$ is small, *e.g.*, $\mu_{\mathcal{E}} = 0, \sigma_{\mathcal{E}} = 0.05$, relatively unimportant edges are likely to be dropped. It verifies that we can control the strength of the augmentation by $\mu_{\mathcal{E}}$ even with the same number of dropped edges. Moreover, by adjusting $\sigma_{\mathcal{E}}$, we can control the diversity of augmentation. The mini map in the fourth column of Figure 1 shows that if $\sigma_{\mathcal{E}}$ increases, the Edge-Drop probability of all edges becomes uniform, *i.e.*, MH-Aug subsumes DropEdge as a special case. Thus, if we set $\sigma_{\mathcal{E}}$ to be large, MH-Aug generates samples with diverse augmentation strength. In sum, the ego-graph perspective enables the explicit control of augmentation strength and diversity to make an advanced augmentation.

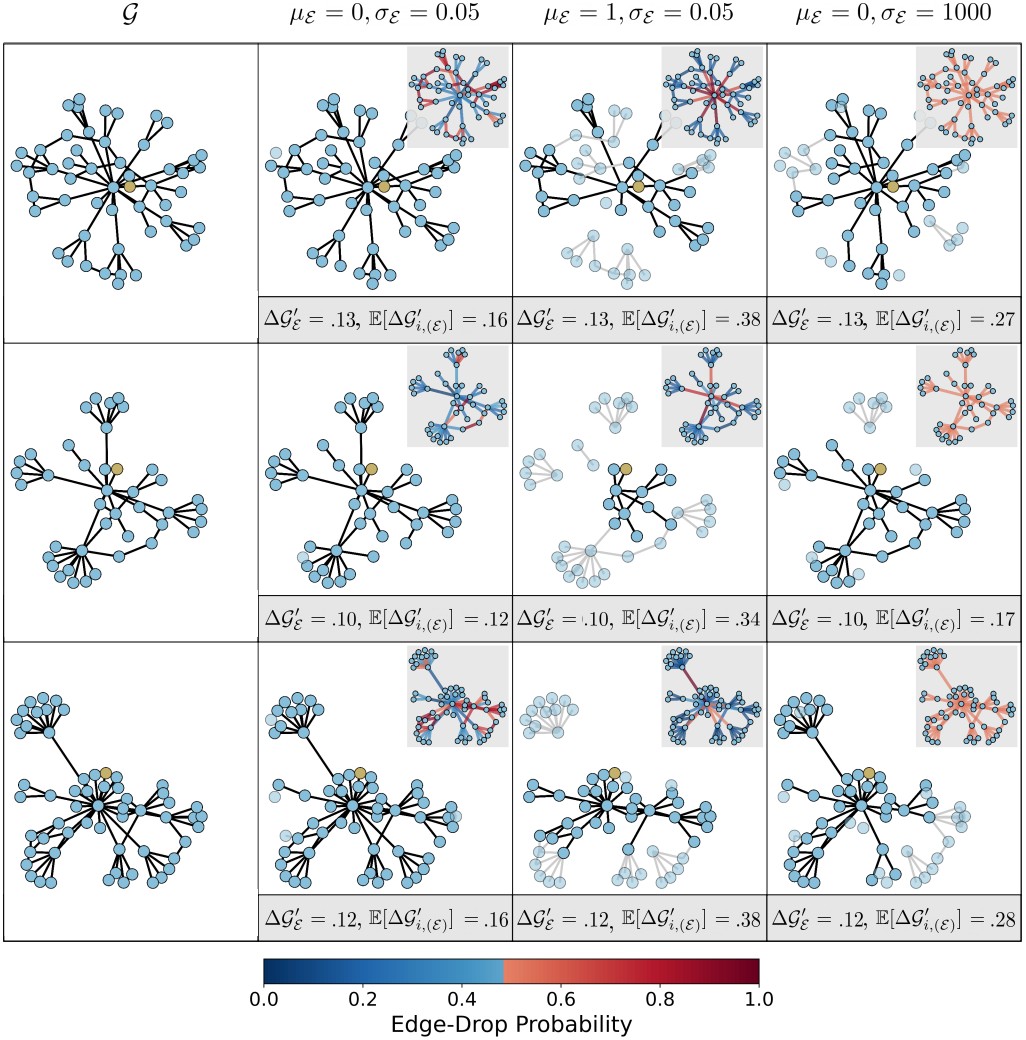

Figure 1: **Diverse $\mathcal{G}'$ sampled by MH-Aug.** Additional visualizations of subgraphs on CORA dataset. The original graphs $\mathcal{G}$ on the first column are 3-hop ego-graphs from yellow nodes. The graphs on other columns are augmented graphs drawn by MH-Aug.

## D.2 Ablation Study

As an ablation study, we conduct experiments with MH-Aug (w/o Reg), our framework trained in the supervised setting, MH-Aug (w/o Normalization), which has the target distribution without normalization term (*i.e.*, $\lambda_2 = 0$ and $\lambda_4 = 0$) and MH-Aug (w/o Entropy), which trained in the semi-supervised setting but without entropy loss $\mathcal{L}_h$. Table 3 shows the contribution of each component in our framework. The results present the importance of each component and it is worth noting that normalization is *crucial* to generate desired samples. On Computers, we observed that MH-Aug (w/o Normalization) underperforms the vanilla supervised training.

## D.3 Results on Large Datasets

We also evaluate our framework on two relatively large datasets – CORA Full and ogbn-arxiv. Cora-Full is proposed in [2] and ogbn-arxiv is in [3]. We randomly select $K \in \{10, 20\}$ nodes per class from the training set. We conduct 10 trials for CORA Full and 3 trials for ogbn-arxiv and report the mean and standard deviation over the trials for every experiment. The results are presented in Table

Table 3: Ablation study of MH-Aug with GCN.

| Methods | CORA | CITESEER | Computers | Photo | CS |
|---|---|---|---|---|---|
| Vanilla | $81.54_{\pm 0.76}$ | $71.64_{\pm 0.31}$ | $79.68_{\pm 2.16}$ | $89.02_{\pm 1.49}$ | $91.45_{\pm 0.28}$ |
| MH-Aug (w/o Reg) | $83.55_{\pm 0.34}$ | $72.96_{\pm 0.48}$ | $80.95_{\pm 2.03}$ | $89.65_{\pm 1.67}$ | $91.81_{\pm 0.33}$ |
| MH-Aug (w/o Normalization) | $82.88_{\pm 0.62}$ | $74.81_{\pm 0.48}$ | $73.49_{\pm 2.20}$ | $90.75_{\pm 1.41}$ | $92.45_{\pm 0.62}$ |
| MH-Aug (w/o Entropy) | $82.53_{\pm 0.49}$ | $71.63_{\pm 0.57}$ | $80.88_{\pm 1.90}$ | $89.82_{\pm 1.63}$ | $91.96_{\pm 0.60}$ |
| MH-Aug | $\mathbf{85.16}_{\pm \mathbf{0.35}}$ | $\mathbf{75.49}_{\pm \mathbf{0.29}}$ | $\mathbf{82.80}_{\pm \mathbf{2.08}}$ | $\mathbf{90.87}_{\pm \mathbf{1.49}}$ | $\mathbf{92.60}_{\pm \mathbf{0.43}}$ |

4. We can observe that MH-Aug outperforms overall base model (*i.e.*, GCN[10], GraphSAGE[11] and GAT[12]). Our framework surpasses the vanilla models on both ogbn-arxiv and CORA Full datasets. When K=10, MH-Aug achieves the improvement of 6.83% against the vanilla model on average. With 10 labeled data per class, our method provides 6.25% performance improvement compared to the vanilla model on average.

Table 4: Node classification results on large datasets with K labeled data per class.

| BaseGNNs | Method | ogbn-arxiv | | CORA Full | |
|---|---|---|---|---|---|
| | | K=10 | K=20 | K=10 | K=20 |
| GCN | Vanilla | $47.83_{\pm 1.13}$ | $54.17_{\pm 1.80}$ | $54.14_{\pm 0.72}$ | $59.99_{\pm 0.39}$ |
| | MH-Aug | $\mathbf{56.19}_{\pm \mathbf{0.44}}$ | $\mathbf{61.22}_{\pm \mathbf{0.33}}$ | $\mathbf{56.27}_{\pm \mathbf{1.97}}$ | $\mathbf{62.79}_{\pm \mathbf{0.49}}$ |
| GraphSAGE | Vanilla | $42.35_{\pm 1.52}$ | $49.19_{\pm 0.42}$ | $52.31_{\pm 1.18}$ | $58.85_{\pm 0.43}$ |
| | MH-Aug | $\mathbf{59.01}_{\pm \mathbf{0.28}}$ | $\mathbf{61.09}_{\pm \mathbf{0.67}}$ | $\mathbf{52.46}_{\pm \mathbf{0.88}}$ | $\mathbf{62.57}_{\pm \mathbf{0.73}}$ |

# E    Further Discussion

## E.1    Normalization Term of Target Distribution

As mentioned in Figure 4 of the main paper, the normalization in the target distribution $P$ by the number of possible augmented graphs corresponding to the same change ratio, $\left(\binom{|\mathcal{E}|}{|\mathcal{E}| \cdot \Delta \mathcal{G}'_{\mathcal{E}}}\right)$, is crucial to generate ego-graphs with the desired ego-graph change ratio $\mu_{\mathcal{E}}$. In this part, we simulate the sampling process of MH-Aug with a simple distribution (*i.e.*, uniform distribution) as the target distribution. To examine the necessity of normalization, we observe the distribution of augmented graphs w.r.t. the *change ratio space*, *i.e.*, $\Delta \mathcal{G}'_{\mathcal{E}}$ (the first row in Figure 2) and the *sample space*, *i.e.*, $\mathcal{G}'$ (the second row in Figure 2). If we define the target distribution without normalization, it follows uniform distribution aspect to the *sample space* as shown in Figure 2(c). However, in the perspective of the *change ratio space*, the change ratio is concentrated on 0.5 as described in Figure 2(a). This is natural, as the number of possible subgraphs on 0.5 change ratio $\left(\binom{|\mathcal{E}|}{|\mathcal{E}| \cdot 0.5}\right)$ is overwhelmingly large compared to the other change ratios. Since our desired target distribution is related to the change ratio $\Delta \mathcal{G}'_{\mathcal{E}}$, we need to generate samples that follow target distribution (*e.g.*, uniform distribution) in terms of the *change ratio space*. By adopting normalization with the number of possible augmented graphs corresponding to the same change ratio, MH-Aug generates augmented graphs following the target distribution in the perspective of the *change ratio space* as shown in Figure 2(b). Note that, if we draw the same number of subgraphs for each change ratio, it is inevitable to sample the identical subgraphs that have extreme change ratios. For instance, Figure 2(d) shows that the original graph ($\Delta \mathcal{G}'_{\mathcal{E}} = 0$) and the graph without any edge ($\Delta \mathcal{G}'_{\mathcal{E}} = 1$) are most frequently sampled since they have only one possible subgraph with the same change ratio.

---

[10]Copyright (c) 2016 Thomas Kipf. Licensed under MIT License

[11]Copyright (c) 2017 William L. Hamilton, Rex Ying. Licensed under MIT License

[12]Copyright (c) 2018 Petar Veličković. Licensed under MIT License

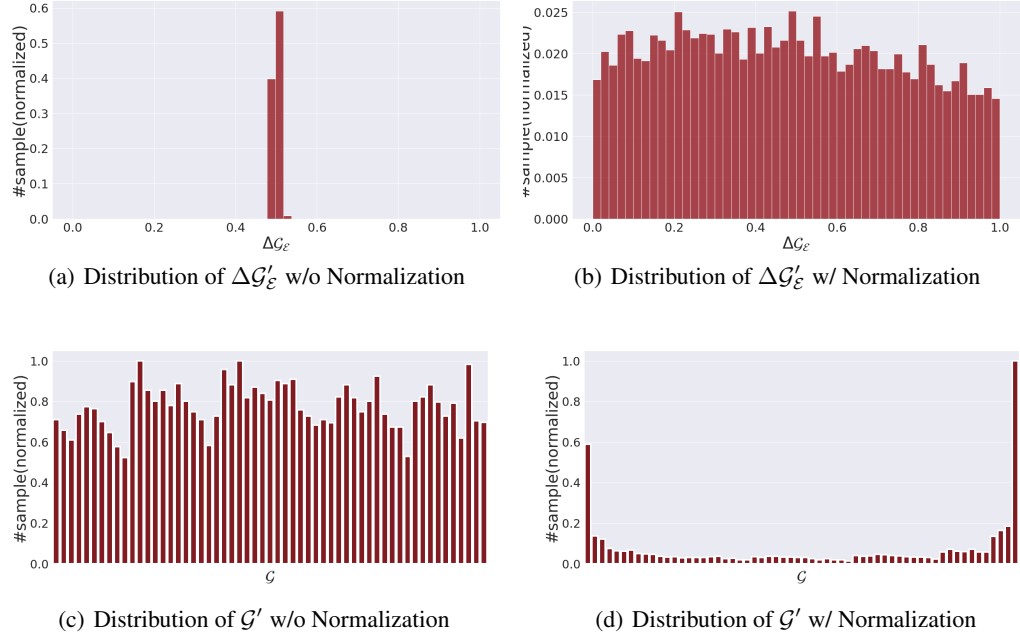

(a) Distribution of $\Delta\mathcal{G}'_{\mathcal{E}}$ w/o Normalization

(b) Distribution of $\Delta\mathcal{G}'_{\mathcal{E}}$ w/ Normalization

(c) Distribution of $\mathcal{G}'$ w/o Normalization

(d) Distribution of $\mathcal{G}'$ w/ Normalization

Figure 2: **The effect of normalization term.** The plots above show the distribution of augmented graphs w.r.t. the *change ratio space* $\Delta\mathcal{G}_{\mathcal{E}}$ and the *sample space* $\mathcal{G}$. We conduct the experiments with a uniform distribution as the target distribution. (a) and (b) present the distribution of the *change ratio* $\Delta\mathcal{G}_{\mathcal{E}}$ w/o and w/ normalization. (c) and (d) show the distribution of *sample* $\mathcal{G}$ w/o and w/ normalization.

## E.2 Calculation of Acceptance Ratio

In Metropolis-Hastings algorithm, the acceptance ratio $\mathcal{A}$ is given as:

$$\mathcal{A} = \min\left\{1, \frac{P(\mathcal{G}')Q(\mathcal{G}^{(t)}|\mathcal{G}')}{P(\mathcal{G}^{(t)})Q(\mathcal{G}'|\mathcal{G}^{(t)})}\right\}. \tag{8}$$

Here, we only consider the augmentation of edge for simplicity. The target distribution of MH-Aug can be written as follow:

$$P_{\mathcal{E}}(\mathcal{G}') \propto \left[\prod_i^{|\mathcal{V}|} \exp\left(-\frac{(\Delta\mathcal{G}'_{i,(\mathcal{E})} - \mu_{\mathcal{E}})^2}{2\sigma_{\mathcal{E}}^2}\right)\right]^{\lambda_1} \cdot \left[\frac{1}{\binom{|\mathcal{E}|}{|\mathcal{E}|\cdot\Delta\mathcal{G}'_{\mathcal{E}}}}\right]^{\lambda_2},$$

$$\ln P_{\mathcal{E}}(\mathcal{G}') \propto \lambda_1 \sum_i^{|\mathcal{V}|}\left[-\frac{(\Delta\mathcal{G}'_{i,(\mathcal{E})} - \mu_{\mathcal{E}})^2}{2\sigma_{\mathcal{E}}^2}\right] - \lambda_2 \ln\binom{|\mathcal{E}|}{|\mathcal{E}|\cdot\Delta\mathcal{G}'_{\mathcal{E}}} \tag{9}$$

The proposal distribution of MH-Aug can be written as:

$$Q_{\mathcal{E}}(\mathcal{G}^{(t)}|\mathcal{G}') \propto \frac{\phi(\xi_{\mathcal{E}}^{(t)})}{\Phi(\beta'_{\mathcal{E}}) - \Phi(\alpha'_{\mathcal{E}})} \cdot \frac{1}{\binom{|\mathcal{E}|}{|\mathcal{E}|\cdot\Delta\mathcal{G}'_{\mathcal{E}}}},$$

$$\ln Q_{\mathcal{E}}(\mathcal{G}^{(t)}|\mathcal{G}') \propto \ln\frac{\phi(\xi_{\mathcal{E}}^{(t)})}{\Phi(\beta'_{\mathcal{E}}) - \Phi(\alpha'_{\mathcal{E}})} - \ln\binom{|\mathcal{E}|}{|\mathcal{E}|\cdot\Delta\mathcal{G}_{\mathcal{E}}^{(t)}}, \tag{10}$$

where $\alpha'_{\mathcal{E}} = \frac{a - \Delta\mathcal{G}'_{\mathcal{E}}}{\sigma_{\Delta}}$, $\beta'_{\mathcal{E}} = \frac{b - \Delta\mathcal{G}'_{\mathcal{E}}}{\sigma_{\Delta}}$, $\xi_{\mathcal{E}}^{(t)} = \frac{\Delta\mathcal{G}_{\mathcal{E}}^{(t)} - \Delta\mathcal{G}'_{\mathcal{E}}}{\sigma_{\Delta}}$, $\phi(x) = \frac{1}{\sqrt{2\pi}}\exp(-\frac{1}{2}x^2)$ as the probability density function of the standard normal distribution and $\Phi(x) = \frac{1}{2}(1 + erf(\frac{x}{\sqrt{2}}))$ as its cumulative distribution function. For simplicity, we denote the first term (*i.e.*, truncated Gaussian given $\mu = \mathcal{G}'$,

$\sigma = \sigma_\Delta$, and range=[a,b]) as $\mathcal{TN}(\mathcal{G}^{(t)}; \mathcal{G}', \sigma_\Delta, a, b)$ where a and b are the range of truncated Gaussian distribution and they are fixed to 0 and 1 respectively. Then, $\alpha = \frac{P(\mathcal{G}')Q(\mathcal{G}^{(t)}|\mathcal{G}')}{P(\mathcal{G}^{(t)})Q(\mathcal{G}'|\mathcal{G}^{(t)})}$ can be calculated as follow:

$$
\begin{aligned}
\ln \alpha &= \ln P(\mathcal{G}') - \ln P(\mathcal{G}^{(t)}) + \ln Q(\mathcal{G}^{(t)}|\mathcal{G}') - \ln Q(\mathcal{G}'|\mathcal{G}^{(t)}) \\
&= \lambda_1 \sum_i^{|\mathcal{V}|} \left[ -\frac{(\Delta\mathcal{G}'_{i,(\mathcal{E})} - \mu_\mathcal{E})^2}{2\sigma_\mathcal{E}^2} \right] - \lambda_2 \ln \binom{|\mathcal{E}|}{|\mathcal{E}| \cdot \Delta\mathcal{G}'_\mathcal{E}} \\
&\quad - \lambda_1 \sum_i^{|\mathcal{V}|} \left[ -\frac{(\Delta\mathcal{G}^{(t)}_{i,(\mathcal{E})} - \mu_\mathcal{E})^2}{2\sigma_\mathcal{E}^2} \right] + \lambda_2 \ln \binom{|\mathcal{E}|}{|\mathcal{E}| \cdot \Delta\mathcal{G}^{(t)}_\mathcal{E}} \\
&\quad + \ln \mathcal{TN}(\mathcal{G}^{(t)}|\mathcal{G}') - \ln \binom{|\mathcal{E}|}{|\mathcal{E}| \cdot \Delta\mathcal{G}^{(t)}_\mathcal{E}} \\
&\quad - \ln \mathcal{TN}(\mathcal{G}'|\mathcal{G}^{(t)}) + \ln \binom{|\mathcal{E}|}{|\mathcal{E}| \cdot \Delta\mathcal{G}'_\mathcal{E}} \\
&= \lambda_1 \sum_i^{|\mathcal{V}|} \left[ -\frac{(\Delta\mathcal{G}'_{i,(\mathcal{E})} - \mu_\mathcal{E})^2}{2\sigma_\mathcal{E}^2} \right] - \lambda_1 \sum_i^{|\mathcal{V}|} \left[ -\frac{(\Delta\mathcal{G}^{(t)}_{i,(\mathcal{E})} - \mu_\mathcal{E})^2}{2\sigma_\mathcal{E}^2} \right] \qquad (11) \\
&\quad + \ln \mathcal{TN}(\mathcal{G}^{(t)}|\mathcal{G}') - \ln \mathcal{TN}(\mathcal{G}'|\mathcal{G}^{(t)}) \\
&\quad + (1-\lambda_2) \ln \binom{|\mathcal{E}|}{|\mathcal{E}| \cdot \Delta\mathcal{G}'_\mathcal{E}} - (1-\lambda_2) \ln \binom{|\mathcal{E}|}{|\mathcal{E}| \cdot |\Delta\mathcal{G}^{(t)}_\mathcal{E}|} \\
&= \lambda_1 \sum_i^{|\mathcal{V}|} \left[ -\frac{(\Delta\mathcal{G}'_{i,(\mathcal{E})} - \mu_\mathcal{E})^2}{2\sigma_\mathcal{E}^2} \right] - \lambda_1 \sum_i^{|\mathcal{V}|} \left[ -\frac{(\Delta\mathcal{G}^{(t)}_{i,(\mathcal{E})} - \mu_\mathcal{E})^2}{2\sigma_\mathcal{E}^2} \right] \\
&\quad + \ln \mathcal{TN}(\mathcal{G}^{(t)}|\mathcal{G}') - \ln \mathcal{TN}(\mathcal{G}'|\mathcal{G}^{(t)}) \\
&\quad + (1-\lambda_2) \ln \frac{Beta(|\mathcal{E}| - |\mathcal{E}| \cdot \Delta\mathcal{G}^{(t)} + 1, |\mathcal{E}| \cdot \Delta\mathcal{G}^{(t)} + 1)}{Beta(|\mathcal{E}| - |\mathcal{E}| \cdot \Delta\mathcal{G}' + 1, |\mathcal{E}| \cdot \Delta\mathcal{G}' + 1)},
\end{aligned}
$$

where $Beta(\cdot)$ is the beta function. Since the calculation of a combination might cause a numerical error, in our implementation we adopt *betaln* in scipy [14] instead of the combination. It can be calculated similarly in the case of considering both edge and node augmentation.

### E.3 Negative Societal Impacts and Limitations

**Negative Societal Impacts.** MH-AUG is a generic semi-supervised learning framework with a novel augmentation method. We believe that there is no direct negative societal impact of this work. However, like other augmentation methods or GNN models for graph-structured data, our generic framework can be used for malicious activities. GNNs are capable to extract uncharted knowledge from the graph data (*e.g.*, a graph with private information extracted from social media). Through the misappropriation of graphs with sensitive information, it might incite or manipulate public opinion. To mitigate the potential problems, illegal data collection should be prevented and it is encouraged to put the dataset under rigorous scrutiny before the release.

**Limitations.** Since we adopt the MCMC sampling for augmentation, our method also has the burn-in issue that is common in MCMC methods. To mitigate this problem, we start from the original graph which is at an area with a high probability density. But, for an extreme target distribution, this initialization might not be sufficient to resolve the burn-in issue and better initialization needs to be studied. In addition, we propose a simple target distribution based on Gaussian distribution with a normalization. However, there might be more ideal target distributions for effective augmentation. These are left for future works.