# OpenReview forum: "Metropolis-Hastings Data Augmentation for Graph Neural Networks"
_NeurIPS.cc/2021/Conference — NeurIPS 2021 Poster_

### Official Review · Reviewer_S7LY · 2021-07-09

**Rating:** 6
**Confidence:** 4

**Summary:**

This paper proposes a data augmentation method for GNNs based on Metropolis-Hastings (MH) sampling  algorithm. The technical method is novel and shown to be effective empirically.

**Limitations And Societal Impact:**

Yes

**Main Review:**


Based on MH algorithm, this paper provides a data augmentation method for training GNNs under semi-supervised setting. Overall, the technical contribution is clear and the empirical study can demonstrate the effectiveness of the proposal.


#######

Pros:

(1) This work forms the data augmentation for graphs as a MCMC sampling process, which is new and potentially insightful to the community.

(2) To utilize MH algorithm for sampling, this work defines reasonable target distribution and proposal distribution, which are the key points that make the proposal work.

(3) The experimental results can show the effectiveness of the proposed data augmentation method. The ablation studies and the visualizations are pretty good for understanding.

########

Cons:

(1) The proposed method is a MCMC-based method, which is assumed to be time-consuming. However, there is no theoretical or/and empirical analysis of the efficiency of the proposed method.

(2) The proposed regularizer terms, especially the term $L_u$, are quite interesting. But the description for this is very limited. For example, since $L_u$ includes model output from both $t$ and $t+1$ steps, more explanation on how to do optimization/back-propagation is necessary.

**Time Spent Reviewing:**

3

---

> ### Author Response · Authors · 2021-08-10
> **Initial Response to R3**
>
> We appreciate your positive comments on our novelty and constructive detailed feedback. We will address every issues raised and reflect on the final paper.
> ***
> **Question1:** The proposed method is a MCMC-based method, which is assumed to be time-consuming. However, there is no theoretical or/and empirical analysis of the efficiency of the proposed method.
>
> **Answer:** We conduct an empirical analysis to compare the time efficiency with other baselines. We use GCN as the base model and average the ten training times for 100 epochs on a single GPU (Tesla V100). Table 1 shows the comparison results between MH-Aug and the baselines. Note that at inference time, there exists no overhead in our method since MCMC sampling is only performed for training time.
>
> |	Method |	CORA |	CITESEER |	Computers |	Photo |	CS |
> |-|-|-|-|-|-|
> |	vanilla|	1.26 ± 0.03 |	1.18 ± 0.04|	1.94 ± 0.02|	1.49 ± 0.02|	2.80 ± 0.02|
> | DropEdge|	1.29 ± 0.03|	1.20 ± 0.04|	1.99 ± 0.18|	1.49 ± 0.03|	2.79 ± 0.02|
> | *AdaEdge*|	 0.36 ± 0.02 (13.72*)|0.42 ± 0.05 (21.46*) |0.76 ± 0.08 (364.17*) | 0.47 ± 0.08 (119.82*)| 1.31 ± 0.07  (640.09*)|
> |	 *GAug-M*|	 0.38 ± 0.09 (29.25*)|	 0.39 ± 0.08 (47.31*)|	 0.77 ± 0.11 (727.83*)|	0.50 ± 0.07 (249.02*) |	1.40 ± 0.08 (1117.03*) |
> |	 *GAug-O*|	4.02 ± 0.09|	7.10 ± 0.03|	OOM|	56.54 ± 0.13|	OOM|
> |	**MH-Aug**|	3.04 ± 0.15|	2.95 ± 0.37|	3.77 ± 0.17|	2.56 ± 0.08|	5.82 ± 0.19|
>
> Table 1. Comparisons of the computation time (s) between MH-Aug and other baselines.
> > \* denotes preprocessing/pretraining time (s) before training
>
> When drawing a sample in the Metropolis-Hastings algorithm, it often rejects candidate graphs, thereby MH-Aug takes longer compared to the vanilla training. However, we observe that MH-Aug is efficient compared to other learning-based augmentations (e.g., AdaEdge and GAug) that require additional preprocessing or pretraining steps which cause the computational overhead. In particular, on CS dataset, AdaEdge takes 1.31 seconds for 100 epochs, plus an additional preprocessing time of 640.09 seconds. GAug-M takes 1.40 seconds for 100 epochs, plus an additional pretraining time of 1117.03 seconds. In contrast, MH-Aug only takes 5.82 seconds on the same dataset. Even for the smallest dataset CORA, AdaEdge and GAug-M take overall 4~10 times more than our method. We thank you for bringing this issue to our attention and we will provide the empirical analysis in the supplementary material.
> ***
> **Question2:** The proposed regularizer terms, especially the term L_u, are quite interesting. But the description for this is very limited. For example, since L_u includes model output from both t and  t+1 steps, more explanation on how to do optimization/back-propagation is necessary.
>
> **Answer:** Great question! In Equation 8, $\mathcal{L}_u$ is the regularizer for unlabeled data to constrain predictions to be invariant to input noise [1]. As stated in Line 176-182, $\mathcal{L}_u$ encourages the consistency of predictions on two consecutive augmented samples $\mathcal{G}_i^{(t)}$ and $\mathcal{G}_i^{(t+1)}$.
>
> In $\mathcal{L}_u$, the model parameter $\theta$ is not fixed for both different augmented graphs in $t$ and $t+1$ steps. It means that we perform back-propagation with both $\mathcal{G}_i^{(t)}$ and $\mathcal{G}_i^{(t+1)}$. We are aware that the stop gradient is commonly used in consistency training [1] and self-supervised learning [2], but our preliminary experiment showed that there was no significant performance gain. So we optimize the model parameter $\theta$ for both $\mathcal{G}_i^{(t)}$ and $\mathcal{G}_i^{(t+1)}$.
>
> [1] Qizhe Xie, Zihang Dai, Eduard Hovy, Thang Luong, and Quoc Le. Unsupervised data augmentation for consistency training. In NeurIPS, volume 33, 2020.
>
> [2] Sohn, Kihyuk and Berthelot, David and Li, Chun-Liang and Zhang, Zizhao and Carlini, Nicholas and Cubuk, Ekin D and Kurakin, Alex and Zhang, Han and Raffel, Colin. Fixmatch: Simplifying semi-supervised learning with consistency and confidence. In NeurIPS, volume 33, 2020

---

> > ### Comment · Reviewer_S7LY · 2021-08-25
> > **Thanks for response**
> >
> > Thank you for your response. The response solves my questions well. Please consider including this information in the next version.

---

### Official Review · Reviewer_1YEx · 2021-07-14

**Rating:** 6
**Confidence:** 4

**Summary:**

The paper proposes a new data augmentation method for Graph Neural Networks based on Markov Chain Monte Carlo sampling and in particular the Metropolis-Hastings algorithm in the context of semi-supervised learning. The main intuition of the paper is to define a target distribution for the whole graph which is representative of the changes in the number of messages that are exchanged in the k-hop neighborhood of each node and then apply the Metropolis-Hastings algorithm for obtaining samples that are representative of this distribution. The experimental section compares the proposed method (MH-Aug) to other recent data augmentation methods showing favorable results w.r.t. the baselines.

**Limitations And Societal Impact:**

Yes

**Main Review:**

The paper proposes an interesting idea for data augmentation for graph structured data. To the best of my knowledge, this is the first time that the Metropolis-Hastings algorithm is used for sampling variations of an input graph, whose strength can be controlled via hyper-parameters for improving performance of graph neural networks. The experimental section of the paper shows good results for the proposed method, which outperforms the selected baselines in all the considered prediction tasks.

While I believe the idea presented in the paper is intriguing (and the task for sure of relevance to the community), the paper itself is not particularly well written. The introduction on the Metropolis-Hastings algorithm is rather short and high level and assumes that the reader is already familiar with the details of the method (e.g. the acceptance score of equation 7 is not explained in the paper and it’s given for granted that the reader is knowledgeable of the algorithm). Throughout the paper I personally found confusing the continuous jumps between changes of measures defined at an ego-network and graph level (e.g. equations 2 and 3 are defined for the whole graph G in terms of variations of the ego networks of the nodes of G which are presented after the equations themselves; the sampling function Q of equation 6 is instead defined resorting to the variations of the number of edges / nodes of whole G, which are presented before equation 2 and 3 - reading this section personally felt like reading spaghetti code, where I had to continuously go back and forth to clarify the intent of the authors). Additionally, there are a number of missing definitions that make the understanding of the whole paper rather difficult, in particular:

1) Lines 128/130, this is very unclear, the entropy of the prediction is a scalar at node i, what do the authors mean with standard deviation of the entropy of a target node here?

2) In equation 2 the expected variation is defined wrt the entropy \epsilon_i of the predictions of node i, but this is not defined in the text, what do the authors mean here?

3) Line 164, a and b I believe represent the extremes of the truncated gaussian, however this is not defined at all in the main paper (there are details of this in the supplementary material, but I believe the authors should not rely on that for these sort of fundamental definitions that build the main message of the paper).

4) Lines 177/178, the whole framework is targeted towards semi-supervised learning as the whole graph is modified by equation 6 instead of the single independent ego networks (unless I’m mistaken). Referring to two different training settings, one of which is named “supervised” in the paper is confusing and I would encourage the authors to refactor those lines accordingly.

5) In Figure 3 it’s unclear the meaning of E(\Delta G_i), I believe the authors might intend with this the expected value of \Delta G_i over all possible nodes of the ego network, however this is not detailed in the paper (and if this is correct, I would encourage them to use the classic notation E[.]).

6) Line 237, the meaning of MH-Aug (w/o Reg) is unclear from the paper. Are the authors training on the augmented samples as if they were the original graph here? Please clarify.

7) Lines 253/254, the authors mention the edge drop probabilities are computed from equation 2, however equation 2 determines the similarity of the ego networks w.r.t. the desired variation rather than the probability of an edge being dropped. I would ask the authors to provide more details on this and clarify the computation of those probabilities in the rebuttal.

8) Lines 275/282, the references to Figure 5a and 5b should be swapped.

In light of this, I’m weakly in favor of the acceptance of the paper and I would ask the authors to clarify all the doubts addressed above and refactor the paper accordingly to improve readability.


**Time Spent Reviewing:**

6

---

> ### Author Response · Authors · 2021-08-10
> **Initial Response to R2**
>
> We appreciate your acknowledgement on our novelty and constructive feedback. We will address every issue raised and reflect on the final paper.
>
> ***
>
> **Question1)** Shortage of explanation regarding the Metropolis-Hastings algorithm.
>
> **Answer:** We appreciate for pointing out this issue and agree some readers may not be familiar with the Metropolis-Hastings algorithm. We will provide basic concepts of the Metropolis-Hastings algorithm in detail.
>
> ***
>
> **Question2)**  Reorganizing the order of descriptions on change ratios defined at different levels (e.g., whole graph level and ego-graphs level).
>
> **Answer:** We apologize for the confusion caused by the order of descriptions regarding different change ratios in the Section 3 of the main paper.  As your suggestion, we will reorganize the structure to enhance the readability in the final version. For calculating the proposal distribution $Q\left(\mathcal{G}^{\prime} \mid \mathcal{G}^{(t)}\right)$ and the normalization term in the target distribution $P(\mathcal{G'})$, we need change ratios at the whole graph level ($\Delta{\mathcal{G'\_E}},\Delta{\mathcal{G'\_V}}$). As such, for calculating the target distribution $P(\mathcal{G'})$, change ratios at the ego-graph graph level are requisite ($\Delta \mathcal{G}\_{i,(\mathcal{E})}^{\prime}, \Delta \mathcal{G}_{i,(\mathcal{V})}^{\prime}$). Since we adopt the well-known approaches for the former, i.e., $\Delta{\mathcal{G'\_E}} = 1 - {|\mathcal{E}'|\over|\mathcal{E}|}, \Delta{\mathcal{G'\_V}} = 1 - {|\mathcal{V}'|\over|\mathcal{V}|}$, we first introduced them in the preliminary section. On the contrary, to highlight our proposed method that makes the calculation time and memory efficient using the structure of ego-graphs, we introduced the latter after delineating the $P(\mathcal{G'})$.
>
> ***
>
> **Question3)** Addressing unclear, missing descriptions, typos, and minor fixes.
>
> **Answer:** We are grateful for your detailed feedback on unclear description, missing definition, typos and unclear notations. We will address them all below and incorporate your feedback in the final version.
>
> ***
>
> **(Q3.1, Q3.2)** Clarifying **$\mu_{\mathcal{E}}(\cdot)$** and $\sigma_{\mathcal{E}}(\cdot)$ in Equation 2 of the main paper.
>
> **Answer:** We apologize for the lack of explanation on **$\mu_{\mathcal{E}}(\cdot)$** and $\sigma_{\mathcal{E}}(\cdot)$ in Equation 2 of the main paper. As we mentioned the definition of standard deviation function $\sigma_{\mathcal{E}}(\cdot)$ in Line 129-130 of the main paper, the mean function $\mu_{\mathcal{E}}(\cdot)$ is also a linear function of the entropy $\epsilon_i$, which is the prediction at node $v_i$. In other words, they can be written as $\mu_{\mathcal{E}}(\epsilon_i) = a\epsilon_i + b$ and $\sigma_{\mathcal{E}}(\epsilon_i) = c\epsilon_i + d$, where a,b,c,d are hyperparameters.
>
> Although $\mu_{\mathcal{E}}$ and $\sigma_{\mathcal{E}}$ could also be adopted as the constant for all the samples, we make them dependent on the entropy of prediction at node $i$, so that the strength and the diversity of augmentation can adaptively vary depending on how confident the model is in its prediction.
>
> ***
>
> **(Q3.3)** Missing the extremes of the truncated Gaussian (e.g., $a$ and $b$ in Line 164) in the main paper.
>
> **Answer:** As you mentioned, $a$ and $b$ in Line 164 are indeed the extremes of the truncated Gaussian, where we provide the detailed definition in the supplement. We will add them in the main paper.
>
> ***
>
> **(Q3.4, Q3.6a)** Clarification on "w/o Reg" vs. "w/ Reg" and "Supervised setting" vs. "Semi-supervised setting".
>
> **Answer:** We have already addressed this issue with the comment (R1-Question1) above. For the reviewer's convenience, we here repeat the answer. As mentioned in Line 216-218 of the main paper, the "supervised" setting means training the model only with the labeled data and cross-entropy loss $\mathcal{L_s}$ whereas the "semi-supervised" setting means using extra regularization losses to explicitly utilize the unlabeled data. In our case, we respectively denote the supervised setting as "w/o Reg" and the semi-supervised setting as "w/ Reg". In addition, since both DropEdge and AdaEdge only use the cross-entropy loss in training, we separate them from other baselines and indicate them as "supervised" settings.
>
> ***
>
> **(Q3.5)** Precise description and notation for $\text{E}(\Delta \mathcal{G}\_{i,(\mathcal{E})})$ in Figure 3 of the main paper.
>
> **Answer:** As you suggested, $\text{E}(\Delta \mathcal{G}\_{i,(\mathcal{E})})$ in Figure 3 is the expected value of $\Delta \mathcal{G}\_{i,(\mathcal{E})}$ concerning all possible nodes of the ego graph. We will specify the details and also change it to the classic notation $\mathbb{E}[\Delta \mathcal{G}\_{i,(\mathcal{E})}]$.
>
> ***
>
> **(Q3.6b)** Are the authors training on the augmented samples as if they were the original graph here?
>
> **Answer:** Yes, we train the model with augmented graphs as if they were the original graph for both supervised and semi-supervised settings.
>
> ***
>
> **(Q3.7)** Computation details on edge-drop probability about Figure 3 described in Line 253-254.
>
> **Answer:** Great Question! We will explain the computation detail of edge-drop probability described in Line 253-254 about Figure 3. In Equation 2, we get the likelihood of augmented graph $P(\mathcal{G'\_E})$, given the desired variation (i.e., $\mu\_{\mathcal{E}}$ and $\sigma\_{\mathcal{E}}$) of ego-graphs. Then, with $P(\mathcal{G'\_E})$, we compute the marginal probability of edge being dropped, i.e., $P(e_j=0)$, which can be written as follows:
>
> $$\large{\displaylines{\begin{aligned}
> P(e\_j=0)
> &= \underbrace{\sum\_{\mathcal{G\_E'} \in \mathfrak{G}\_\mathcal{E}(\mathcal{G},r)}}\_{\text{(1)}}
>  P({e\_j=0}| \mathcal{G}\_\mathcal{E}') P(\mathcal{G\_E'}) \\\
> &= \underbrace{\sum\_{\mathcal{G\_E'}} \mathbb{1}(e\_j \notin \mathcal{G\_E'}) \overbrace{ \left\\{{Z}^{-1} \prod\_{i}^{|\mathcal{V}|} \exp{\left( - {(\Delta \mathcal{G}\_{i, (\mathcal{E})}' - \mu\_\mathcal{E} )^2 \over 2(\sigma\_\mathcal{E})^2 } \right)}\right\\} }\^{\text{(2)}} }\_{\text{(3)}}  ,
> \end{aligned}}}$$
>
>
> where $\mathbb{1}(\cdot)$ is the indicator function and $\mathfrak{G}(\mathcal{G},r)$ is the set of all possible subgraphs with the change ratio $r$  w.r.t $\mathcal{E}$. In other words, $\mathfrak{G}\_{\mathcal{E}}(\mathcal{G}=(\mathcal{V},\mathcal{E}),r)=\\{ \mathcal{G'}=(\mathcal{V},\mathcal{E'}) : {|\mathcal{E'}|\over|\mathcal{E}|}=1-r \ \text{and } \mathcal{E'} \subset \mathcal{E} \\}$,  $|\mathfrak{G}\_{\mathcal{E}}(\mathcal{G},r)|= {|\mathcal{E}|\choose |\mathcal{E}| \cdot r}$  , and $Z$ is the normalization constant where $Z = \sum_{\mathcal{\mathcal{G}\_{\mathcal{E}}'}} \prod_{i}^{|\mathcal{V}|} \exp{\left( - {(\Delta \mathcal{G}'\_{i, (\mathcal{E})}- \mu\_{\mathcal{E}} )^2 \over 2(\sigma\_{\mathcal{E}})^2 } \right)}$. Specifically, we calculate the edge-drop probability $P(e_j=0)$ in 3 steps below:
>
> (1) With the fixed change ratio of edge, $r$ (e.g., $\Delta \mathcal{G\_{E}'}=0.1$), we find all ${|\mathcal{E}|\choose |\mathcal{E}| \cdot r}$   possible subgraphs that have $(1 - r)|\mathcal{E}|$ edges.
>
> (2) For all ${|\mathcal{E}|\choose |\mathcal{E}| \cdot r}$   possible subgraphs, we compute the probabilities of each subgraph $P(\mathcal{G\_{E}'})$ by normalizing the likelihood with $Z$.
>
> (3) Finally, compute the edge-drop probability $P(e_j=0)$ by summing up the probabilities of subgraphs $\mathcal{G'\_{E}}$ only when the edge $e_j$ does not exist in $\mathcal{G'\_{E}}$.
>
> It is worth noting that in Line 254 (for Figure 3 in the main paper), the second term in Equation 2 is canceled out since we have the constant whole graph change ratio $r$. We fixed $r$ for ease of visualization.
>
> ***
>
> **(Q3.8)** Lines 275/282, the references to Figure 5a and 5b should be swapped.
>
> **Answer:** We will correct the reference in Line 275-282 for Figure 5a and 5b at the main paper as you suggested.

---

> > ### Author Response · Authors · 2021-09-01
> > **Gentle Reminder for R2.**
> >
> > Thank you for your time and efforts in reviewing our paper. We have responded to your comments and we believe that most of your suggestions have been resolved. Could you please go over our responses and let us know if you have any further concerns or questions?
> >
> > Thanks, Authors

---

> > > ### Comment · Reviewer_1YEx · 2021-09-01
> > > **Thank you for your answers**
> > >
> > > Dear authors,
> > > thank you for answers, there are no further questions from my side.

---

### Official Review · Reviewer_kMJz · 2021-07-16

**Rating:** 6
**Confidence:** 2

**Summary:**

This paper introduces a MH-style data augmentation to boost the generalization ability of the existing GNNs by generating sequence of augmented graphs. The generation is proven to be converging to the desired target distribution. The experimental results shows a promising result for applying augmented graph during the GNN training.

**Limitations And Societal Impact:**

Yes.

**Main Review:**

Strength:
1. The paper is easy to follow and proposed methods are both theoretically and empirical sound from the results.

Weakness & Questions:
1. What's the difference of w/o and w/ Reg? Is the only difference on whether have two terms in equation 9? For the DropEdge and AdaEdge baseline, why do you call them "supervision"?
2. What's the main intuition behind the given target distribution at Equation (2)?


**Time Spent Reviewing:**

1.5

---

> ### Author Response · Authors · 2021-08-10
> **Initial Response to R1**
>
> We are thankful for your supportive comments on the quality of writing, theoretical analysis and good performance, and insightful questions. We will address all issues below.
> ***
> **Question1:** What's the difference of w/o and w/ Reg? Is the only difference on whether have two terms in equation 9? For the DropEdge and AdaEdge baseline, why do you call them "supervision"?
>
> **Answer:** We apologize for the vague descriptions on few terms. As mentioned in Line 216-218 of the main paper, the "supervised" setting means training the model only with the labeled data and cross-entropy loss $\mathcal{L}_s$ whereas the "semi-supervised" setting means using extra regularization losses to explicitly utilize the unlabeled data. In our case, we respectively denote the supervised setting as "w/o Reg" and the semi-supervised setting as "w/ Reg". In addition, since both DropEdge and AdaEdge only use the cross-entropy loss in training, we separate them from other baselines and indicate them as "supervised" settings.
> ***
> **Question2:** What's the main intuition behind the given target distribution at Equation (2)?
>
> **Answer:** To control the **strength** and **diversity** of augmentation, we define the target distribution of the augmented graph $\mathcal{G'}$ in terms of the ego-graphs $\mathcal{G_{i}'}$ as described in Line 123-130 and Equation 2 in the main paper. The **strength** of the augmentation is measured by the average change ratio of ego-graphs $\Delta \mathcal{G_{i,(\cdot)}'}$ w.r.t. the edges and the nodes. In addition, we define the **diversity** of augmentation as the variability of strength $\Delta \mathcal{G_{i,(\cdot)}'}$. Thus, we assume that the change ratio of each ego-graph $\Delta \mathcal{G_{i,(\cdot)}'}$ follows the Gaussian distribution given the desired variation, the mean $\mu$  for the strength and the standard deviation $\sigma$ for the diversity.
>
> In equation 2, the strength and diversity of the change ratio is controlled by the mean function $\mu_{\mathcal{E}}(\cdot)$ and the standard deviation function $\sigma_{\mathcal{E}}(\cdot)$, which are the simple linear function of the entropy $\epsilon_i$ of prediction at node $v_i$, i.e., $\mu_{\mathcal{E}}(\epsilon_i) = a\epsilon_i + b$ and $\sigma_{\mathcal{E}}(\epsilon_i) = c\epsilon_i + d$, where $a,b,c,$ $d$ are hyperparameters.
>
> Furthermore, as mentioned in Line 134-137 of the main paper and Section F.1 in the supplementary material, it is difficult to generate augmented samples with a low (or high) whole graph change ratio. For instance, when $\Delta \mathcal{G_{\mathcal{E}}'}=0.01$  (or $\Delta \mathcal{G_{\mathcal{E}}'}=0.99$), the number of possible subgraphs on low/high change ratio becomes extremely small compared to the change ratio of $0.5$, i.e., ${|\mathcal{E}|\choose|\mathcal{E}| \cdot 0.01} \ll{|\mathcal{E}|\choose|\mathcal{E}| \cdot 0.5}$ . Therefore, we normalize the target distribution by the number of possible augmented graphs corresponding to the change ratio ${|\mathcal{E}|\choose|\mathcal{E}|\cdot \Delta \mathcal{G_E'}}$.

---

> > ### Comment · Reviewer_kMJz · 2021-08-28
> > **Thanks for response.**
> >
> > Thanks for addressing my questions. I have no more questions at this point.

---

### Decision · Program_Chairs · 2021-09-27

**Decision:**

Accept (Poster)

**Comment:**

This paper proposed to do graph data augmentation with a carefully designed target distribution and the corresponding proposal distribution. The experiments show that the proposed augmentation method can be more effective than existing baselines. During the rebuttal period, all the reviewers found that the authors have effectively resolved their concerns, and are generally positive about the paper. Personally I also like the design of the target distribution which is the key insight of the paper. However none of us are super excited about the outcome, partially due to the significance of the contribution and the potential implication for more general settings. Nevertheless, this paper is a sound contribution in the current scope. No matter of the outcome, we highly encourage the authors to incorporate the additional experiments and clarifications in to the revision of the paper to make it more solid.